

# The Atmospheric Oxidizing Capacity in China:
# Part 2. Sensitivity to emissions of primary pollutants

Jianing Dai[a], Guy P. Brasseur[a,e,f], Mihalis Vrekoussis[b,g,h], Maria Kanakidou[b,d], Kun Qu[b], Yijuan Zhang[b], Hongliang Zhang[c], Tao Wang[f]

10    [a] Environmental Modelling Group, Max Planck Institute for Meteorology, Hamburg, 20146, Germany
      [b] Institute of Environmental Physics (IUP), University of Bremen, Bremen, 28359, Germany
      [c] Department of Environmental Science and Engineering, Fudan University, Shanghai, 200433, China
15    [d] Environmental Chemical Processes Laboratory, Department of Chemistry, University of Crete, Heraklion, 70013, Greece
      [e] National Center for Atmospheric Research, Boulder, Colorado, 80307, USA
      [f] Department of Civil and Environmental Engineering, The Hong Kong Polytechnic University, Hong Kong, China
20    [g] Center of Marine Environmental Sciences (MARUM), University of Bremen, Bremen, 28359, Germany
      [h] Climate and Atmosphere Research Center (CARE-C), The Cyprus Institute, Nicosia, Cyprus

*Correspondence to*: Guy P. Brasseur (guy.brasseur@mpimet.mpg.de)



## Abstract

The Atmospheric Oxidation Capacity (*AOC*), often referred to as the self-cleansing ability of the atmosphere, considerably affects the concentrations of photochemical air pollutants. Despite substantial reductions in anthropogenic emissions of key chemical compounds in China, the mechanisms that determine the changes in the atmospheric oxidation capacity are still not sufficiently understood. Here, a regional chemical transport model is employed to

quantify the sensitivity of air pollutants and photochemical parameters to specified emission reductions in China for conditions of January and July 2018 as representative. The model simulations show that, in winter, a 50% decrease in nitrogen oxides ($NO_x$) emissions leads to an 8-10 ppbv (15-20%) increase in surface ozone concentrations across China. In summer, the ozone concentration decreases by 2-8 ppbv (3-12%) in $NO_x$-limited areas, while ozone

increases by up to 12 ppbv (15%) in volatile organic compounds (VOCs)-limited areas. This ozone increase is associated with a reduced $NO_x$-titration effect and higher levels of hydroperoxyl ($HO_2$) radical due to decreased aerosol uptake. With an additional 50% reduction in anthropogenic VOCs emission, the predicted ozone concentration decreases by 5-12 ppbv (6-15%) in the entire geographic area of China, with an exception in the areas, where the role

of BVOCs is crucial to ozone formation. Further, the adopted reduction in $NO_x$ emission leads to an increase of *AOC* by 18% in VOC-limited areas. This specific increase is associated with the combined effect of enhanced radical cycles associated with the photolysis of oxidized VOCs (OVOCs) and the oxidation of alkenes by hydroxyl (OH) radical and $O_3$. A large reduction of daytime *AOC* in summer results from the reduction in anthropogenic VOCs

emission, with a dominant contribution from the reaction of OH radical with reduced alkenes, followed by the reactions with depleted aromatics and OVOCs. This study highlights that photolysis of OVOCs and oxidation of alkenes in urban areas when $NO_x$ emission is reduced leads to an increase in $O_3$. To mitigate ozone rises in urban areas, a joint reduction in the emission of $NO_x$ and specific VOCs species, including alkenes and aromatics and

photodegradable OVOCs, should be implemented.

Keywords: ozone pollution, emission reduction, nitrogen chemistry, *AOC*






## 1. Introduction

To effectively reduce air pollution in China, the government of the country has implemented stringent actions between 2013 and 2020 (Liu et al., 2020; Liu et al., 2023). In the initial phase, from 2013 to 2017, the control of primary pollutants was particularly effective, with a dramatic decrease in the anthropogenic emissions of fine particles ($PM_{2.5}$), sulfur dioxide ($SO_2$), and nitrogen oxides ($NO_x$) (Liu et al., 2020). After that, a sustained reduction in the emission of $SO_2$, $NO_x$, and $PM_{2.5}$ was achieved with continuous emission control from 2018 to 2020 (Liu et al., 2023). The implementation of the emission control policies has greatly improved China's air quality. However, a significant increase in the surface ozone ($O_3$) concentration was observed from 2013 to 2019, with the rising trend slowing down from 2020 to 2021, but rebounding in 2022 (Liu et al., 2023; China Air 2023). Some studies have documented the explanations for the significantly increasing trend in the surface $O_3$ concentration, including the reduction of $NO_x$ emissions and atmospheric aerosol loading (Li et al., 2019a; Liu et al., 2020). During and after the recent COVID-19 lockdown, ozone pollution has also been reported to happen, which is believed to be favored by the sharp reduction of $NO_x$ and high emissions of VOCs (Li et al., 2021). Looking through these changes over the past decade, we can learn that rapid reductions of emissions may lead to varied ozone chemistry and, thereby, complex changes in ozone concentrations in China.

The response of ozone to reduced $NO_x$ emissions varies with the local photochemical environment and is different in VOC-limited, $NO_x$-limited, or transition regimes (Ou et al., 2016; Dai et al., 2023). In VOC-sensitive regimes, the reduction of $NO_x$ tends to increase ozone formation due to the weakening of NO nitration and the competition between $NO_2$ and VOC for OH radicals (Ou et al., 2016). In $NO_x$-sensitive regimes, $NO_x$ emission reduction decreases the photolysis of $NO_2$, leading to less ozone formation (Ou et al., 2016). Several studies using satellite observations (Wang et al., 2021) and regional models (Zhang S. et al., 2023) have shown that the reduction in anthropogenic emissions has generated a change in the geographical distribution of the ozone formation regimes in China. The shift of ozone sensitivity regimes from VOC-sensitive to transition and/or $NO_x$-sensitive in many metropolitan and suburban regions of East China was also reported by these studies. This shift enables efficient ozone control in $NO_x$-sensitive areas in response to the continuous decrease in $NO_x$ emissions. In VOC-sensitive and transition areas, $NO_x$ emission reduction fails to effectively mitigate ozone pollution, while a coordinated reduction in anthropogenic VOCs (AVOCs) emissions should effectively limit the ozone formation and should therefore be implemented (Liu et al., 2023; Zhang S. et al., 2023). The source of $NO_x$ in VOC-sensitive areas is mainly from fossil fuel combustion, while AVOCs emissions have a range of sources. To establish a cost-effective control over AVOCs emission, the assessment of the contribution of different VOCs species to ozone formation should be accurately estimated for different areas of China.

Aerosol decreases associated with the reduction of primary emissions are expected to continuously affect the effectiveness of ozone control (Liu et al., 2023). Following the



successful controls on anthropogenic emissions since 2013, a substantial reduction in $PM_{2.5}$ concentration was observed in China (Zhai et al., 2019). The reduction in the emission of $NO_x$, $SO_2$, and ammonia ($NH_3$), as the gaseous precursors of secondary inorganic aerosol (SIA, the sum of sulfate ($SO_4^{2-}$), nitrate ($NO_3^-$), and ammonium ($NH_4^+$)), leads to a decrease in the SIA concentration (Meng et al., 2022). The decrease in AVOCs emissions will lead to a decrease

in the concentration of secondary organic aerosol (SOA), given the gas-phase photochemical oxidation of VOCs plays an important role in SOA formation (Yuan et al., 2013; Li et al., 2022). However, the increasing trends of OH radicals will positively influence SOA formation (Wang W. et al., 2022; Wang W. et al., 2023) and provide an increasing portion of the secondary formation in aerosol compositions (An et al., 2019). The interaction of aerosol and

$O_3$ formation has been discussed in many modeling studies (Li et al., 2019; Liu et al., 2020). However, the influences of aerosol on $O_3$ production can be varied due to counteracted aerosol effects and different aerosol concentrations (Tan et al., 2022; Dai et al., 2023). Understanding the changes in aerosol effects on ozone formation when the primary emissions are further reduced is still necessary to implement an efficient air quality control policy.


    Recent observational studies combined with a source apportionment approach using observation-based models have highlighted the role of specific VOCs species, including the alkenes, aromatics, and several OVOCs, in mitigating summertime ozone formation in the urban areas of China (Shi et al., 2023; Wang W. et al., 2022). The notable contributions of

OVOCs to *AOC* as well as the formation of SOA in China have been of concern in many studies (Li et al., 2022; Wang et al., 2023). Since the oxidation of biogenic VOCs (BVOCs) can significantly contribute to the formation of secondary pollutions, the important role of BVOCs in *AOC* and the formation of SOA has also been highlighted in vegetated and highly greening regions in China (Cao et al., 2022; Zhang et al., 2023). However, a comprehensive evaluation

of the influence of different VOCs species on *AOC* and ozone chemistry in different regions of China is still needed. Considering the necessity of implementing coordinated actions among large areas to further alleviate air pollution in China, regional chemical transport models are appropriate tools to assess the quantitative response of various VOCs species and *AOC*-related chemical parameters to emission changes.


    In the companion paper (Part 1; Dai et al., 2023), we used a regional chemical-meteorological model to quantify the relative contribution of different photochemical processes to the formation and destruction of near-surface $RO_x$ and $O_3$ in different chemical environments in China. In Part 2 of the study, with the evaluated model, we assess the response of the photo-

oxidative species and related parameters to the reduction of primary emissions. This paper is structured as follows. Section 2 introduces the setups of the model system and describes the simulations performed for specified reductions in the emissions of primary pollutants. In Section 3, we discuss the changes in the ozone formation regime resulting from emission reductions. We also derive the associated changes in the radical and ozone budgets. Further,

we analyzed the response in the near-surface concentration of ozone precursors, ozone, and aerosols to emission reductions. Finally, we describe the sensitivity of the photochemical parameters and the atmospheric oxidative capacity (*AOC*) to the reduction in emissions. A summary of our study is provided in Sec. 4.





**2. Method**

2.1. Model setting

We use the WRF-Chem model version 4.1.2 (Skamarock et al., 2019), coupled with the gas-
phase chemistry mechanism MOZART and the aerosol module MOSAIC, to simulate the
meteorological fields as well as the transport, the chemical and physical transformations of
trace gases and aerosols. The months of January and July of 2018 were selected as
representative months to conduct the simulations and investigate the changes in secondary
pollution and $AOC$ in response to emission reductions during winter and summer, respectively.
Compared to the standard version of the chemical mechanism, several updates of
heterogeneous uptake over the ambient aerosol were made (Dai et al., 2023). As for SOA
formation in the selected chemical mechanism, the main pathways result from the gas-phase
oxidation of VOCs by atmospheric oxidants (OH, $O_3$, and $NO_3$) and the heterogeneous
formation of glyoxal SOA over the ambient aerosol. The model domain covers the whole
geographical area of China. Analyses of modeling results at eight sites, including four urban
sites (Beijing, Shanghai, Guangzhou, and Chengdu), two rural sites (Wangdu and Heshan), and
two remote sites (Waliguan and Hok Tsui), were also performed in this study. More detailed
information on the model configuration, the model validation, and the sites selected for our
analysis can be found in Part 1 of our paper by Dai et al. (2023).


We used the Multi-resolution Emission Inventory (MEIC v1.3; http://www.meicmodel.org/) to
represent anthropogenic emissions in China and the CAMS-GLOB-ANT v4.2 inventory
(https://eccad.aeris-data.fr/) provided by the Copernicus Atmosphere Monitoring Service
(CAMS) for anthropogenic emissions in the Asian areas outside China. To explore the
sensitivity of secondary pollution and $AOC$ to emission reduction, sensitivity experiments were
designed to separately assess the influence of reduced $NO_x$, AVOCs, and other emissions. As
shown in Table S1, $NO_x$ emissions include the emissions of $NO_2$ and NO, and AVOCs
emissions include these of alkenes ($C_2H_4$, $C_3H_6$, and BIGENE (alkenes with carbon number ≥
4)), alkanes ($C_2H_6$, $C_3H_8$, and BIGALK (alkanes with carbon number ≥ 4)), aromatics (benzene,
toluene, and xylene), alkyne ($C_2H_2$), and OVOCs ($CH_3CHO$, $C_2H_5OH$, $CH_3OH$, $C_{10}H_{16}$,
$CH_3COCH_3$, $CH_2CCH_3CHO$ (MACR), and $CH_2CHCOCH_3$ (MVK)). For other emissions, the
emissions of $NH_3$, $SO_2$, and carbon monoxide (CO) were considered.

2.2. Design of numerical experiment

To explore the sensitivity of secondary pollutants to emissions changes, five numerical
experiments are conducted in both January and July of 2018 (Table 1). In the baseline case,
denoted as "*BASE*", we adopted emissions as described in Sect. 2.1. The concentrations of the
key species modeled in this case have been validated in our companion study (Dai et al., 2023).
To quantify the sensitivity of pollutants to the reduction of $NO_x$ and AVOCs emissions, we
applied arbitrary reductions in the surface emissions of primary pollutants; In the first two
cases, a 50% reduction is applied separately to the baseline $NO_x$ and AVOCs emissions in



whole geographical areas of China. These two sensitivity cases are labeled "*NOx*" and "*AVOCs*", respectively. A third case in which 50% reduction is applied to both $NO_x$ and AVOCs emissions is referred to as "*N+A*". The simulation labeled "*TOTAL*" assumes that all anthropogenic emissions ($NO_x$, AVOCs, and other primary pollutants including CO, $SO_2$, and $NH_3$) are reduced by 50%. The difference between modeled concentrations of pollutants and chemical parameters in the sensitivity cases and the baseline case provides an estimate of the response of secondary pollution and chemistry to different emission reductions. The spatial distribution of the emission fluxes changes for the different cases is shown in Fig. S1.

### 3. Model results

3.1. Changes in ozone regimes

In order to display the impact of emission reduction on the changes in the ozone regimes, we first show the geographical distribution of $NO_x$-limited or VOC-limited ozone formation regimes. We adopt the ratio between the production rate of hydrogen peroxide ($H_2O_2$) and that of nitric acid ($HNO_3$) [$P(H_2O_2)/P(HNO_3)$] as the indicator to distinguish these regimes. An area is assumed to be VOC-limited or $NO_x$-limited if $P(H_2O_2)/P(HNO_3) < 0.06$ or $P(H_2O_2)/P(HNO_3) > 0.2$, respectively (Zhang et al., 2009). The regions with ratios between these two limits represent transition situations (Dai et al., 2023).

Figure 1 shows how the spatial distribution of ozone regimes varies in response to applied emission reductions in both January and July. Under baseline conditions (BASE case), during January, ozone formation in a large part of China, including the north, east, and central areas, as well as the southeastern coastline and PRD regions, is controlled by the availability of VOC (Fig. 1a). In contrast, ozone formation in western China is under the control of $NO_x$, while a small area in southern China is located in transition areas. Generally, ozone regimes change mainly in the southern part of China when emissions are reduced. With a 50% reduction in $NO_x$ emissions (Fig. 1c), transition or VOC-limited regimes in the south and southwest China tends to become $NO_x$-limited for ozone production. With a 50% reduction in AVOCs emissions, some transition areas of southern China are converted to VOC-limited areas (Fig. 1e). With the combined reduction in $NO_x$ and AVOCs emissions (Fig. 1g) as well as in all anthropogenic emissions (Fig. 1i), the VOC-limited regions evolve towards transition or $NO_x$-limited regions in southern China.

During July, for baseline conditions (BASE case) (Fig. 1b), a large fraction of the Chinese territory corresponds to $NO_x$-limited conditions; VOC-limited conditions are found in urban areas including the North China Plain (NCP), the Yangzi River Delta (YRD), the Pearl River Delta (PRD), and the Si Chuan Basin (SCB). The changes in ozone regimes with emission reductions are found mainly in VOC-limited areas and surroundings. With the reduction of $NO_x$ emissions, VOC-limited areas shrink and become more concentrated in a smaller fraction of metropolitan areas (Fig. 1d). With the reduction of AVOCs emissions, VOC-limited areas expand to the surroundings near the metropolitan areas (Fig. 1f). With the combined 50% reduction in the emissions of $NO_x$ and AVOCs (N+A case; Fig. 1h) and of all species (TOTAL





case; Fig. 1j), the calculated change pattern of ozone sensitivity is similar, with a smaller VOC-limited area relative to the BASE case.

At the specific sites examined here (see Sect. 2.1), emission reduction does not modify ozone sensitivity regimes at the urban and rural sites in January (Fig. S2), remaining in VOCs-limited regimes. Except for the remote site Waliguan, at this site, the ozone production is still located in the transition regimes in AVOCs case, while in other cases changing to NOₓ-limited regime. However, the change in ozone regimes is notable in July (Fig. 2). During this season, in the
baseline case, the four urban sites and the rural site of Wangdu are VOC-limited; the Heshan site is located in the transition regime and the two remote sites are located in the NOₓ-limited area. With a 50% reduction in NOₓ emissions, only the Guangzhou site remains in a VOC-limited region, but also with the value of the $P(H_2O_2)/P(HNO_3)$ ratio increasing from 0.008 to 0.045. The other three urban sites shift from VOC-limited regimes to transition areas. With a
50% cut in AVOCs emission, ozone sensitivity at the Hok Tsui site shifts from VOC-limited to transition conditions. If we apply a combined 50% reduction to the emissions of NOₓ and AVOCs (N+A case) and all emissions (TOTAL case), the three sites of Beijing, Chengdu, and Wangdu shift from a VOC-limited to a transition area.

3.2. Changes in the budgets of radicals and ozone

In Part 1 of the present study (Dai et al., 2023), we found that the production of ROₓ radicals in China results primarily from the photolysis of $O_3$, nitrous acid (HONO), and different OVOCs, with a relatively minor role from the ozonolysis of alkenes. Besides, the destruction
of ROₓ radicals results from the termination reactions between different ROₓ radicals and between ROₓ radicals and nitric oxide as well as from the heterogeneous uptake of $HO_2$ on aerosol surfaces. The production rate of odd oxygen ($O_x = O_3 + NO_2$) is associated with recurrent radical reaction chains involving the oxidation of hydrocarbons in the presence of NOₓ. The photochemical destruction of $O_x$ results from several processes, including the
photolysis of $O_3$, followed by the reaction between the electronically excited oxygen atom $O(^1D)$ and water vapor ($H_2O$). Other $O_x$ loss mechanisms include the reactions of ozone with OH, $HO_2$, and different alkenes. In the presence of NOₓ, additional $O_x$ losses are provided by the titration of $O_3$ by NO, followed by the conversion of $NO_2$ to nitric acid ($HNO_3$).

3.2.1. Production and destruction rates of ROₓ

Figure 3 shows the spatial distribution of the changes in the averaged daytime (06:00-19:00 Local Standard Time) production rate of ROₓ radicals ($P(RO_x)$) near the surface resulting from a 50% decrease in NOₓ and AVOCs emissions for conditions representative of January and
July 2018, respectively.

In January, with a 50% reduction in NOₓ emission, the resulting reduction in $P(RO_x)$ is of the order of 0.3-0.7 ppbv h⁻¹ (30-33%) in the urban areas, including in the NCP, YRD, and PRD regions (Fig. 3a). The effect of the reduction in AVOCs emissions on the daytime value of
$P(RO_x)$ is less significant than the effect of NOₓ emission reduction, with decreases of $P(RO_x)$



by only 0.1-0.4 ppbv h$^{-1}$ (10-25 %) in the NCP, YRD and SCB (Fig. 3c), but by up to 0.8 ppbv h$^{-1}$ (35%) in the PRD. More notable effects of AVOCs emission reduction on $P(RO_x)$ in the PRD are mainly attributed to the reduced contribution from the photolysis of OVOCs (see the results at the Heshan and Hok Tsui sites in Fig. 4). Under the combined reduction in NO$_x$ and

AVOCs emissions, the decrease of $P(RO_x)$ ranges within 0.5-0.9 ppbv h$^{-1}$ (30-42%), which is overall larger than the sum of the separated effects of NO$_x$ and AVOCs emission reduction (Fig. 3e). With a further decrease in the emissions of other pollutants, the reduction in the daytime $P(RO_x)$ value, by 0.4-0.8 ppbv h$^{-1}$ (25-35%; Fig. 3g), is slightly lower than the combined effect of NO$_x$ and AVOCs emissions reduction. The weakened effect on $P(RO_x)$ is

related to the higher concentration of HONO, HCHO, non-HCHO OVOCs, and ozone (see Sec. 3.3.1), due to the higher level of oxidants caused by the lower consumption of CO, SO$_2$, and NH$_3$, whose emissions are assumed to be reduced.

In July, the decline of $P(RO_x)$ due to the reduction in NO$_x$ emissions is larger than in winter

(Fig. 3b), with a maximum decrease of 1.2 ppbv h$^{-1}$ (28%) in urbanized areas. The reduction in AVOCs emissions leads to a reduction in $P(RO_x)$ that reaches 0.8 ppbv h$^{-1}$ (18%) in urbanized areas of East China (Fig. 3d). Under the combined reduction of NO$_x$ and AVOCs emissions, the reduced value of $P(RO_x)$ is also larger than when considering their separated effects, which ranges from 0.5 to 1.2 ppbv h$^{-1}$ (12-28%; Fig. 3f). When the reduction is applied to emissions

of other species, the production rate of RO$_x$ decreases by 0.4 to 1.0 ppbv h$^{-1}$ (9-22% Fig. 3h).

Figure 4 shows the averaged value of daytime $P(RO_x)$ at eight monitoring sites in all the cases for January and July, respectively. In January, with the 50% reduced NO$_x$ emissions, the reduction of $P(RO_x)$ is predominantly attributed to the reduced contribution from the photolysis

of HONO at urban and rural sites (larger than 70%), which acts as a source of OH. As HONO mainly emanates from the heterogeneous conversion of NO$_2$ in polluted East China (Zhang et al., 2021), the reduced contribution from HONO is associated with the reduced concentrations of NO$_2$. In July, the reduced contribution from the HONO photolysis (due to reduced NO$_x$ emissions) is also an important factor in explaining the decrease in the $P(RO_x)$ values (larger

than 70%) in urban areas. When considering the decrease in AVOCs emissions, the reduced value in daytime $P(RO_x)$ results to a large extent from the reduction in the photolysis rates of OVOCs in both seasons.

Figure S3 shows the spatial distribution of the average daytime changes in the destruction rate

of RO$_x$ radicals ($D(RO_x)$) due to the emission reductions applied in January and July conditions. The reduction in NO$_x$ emissions has a higher effect on $D(RO_x)$ than the reduction in AVOCs emissions. The dominant reason for the decrease in the value of $D(RO_x)$ is due to the reduced destruction rate of OH by nitrogen oxides (NO$_2$ and NO) in January and July (Fig. S4). A higher contribution from the HO$_2$ uptake by aerosol, increasing from 3-5% to 5-10%

in January and July, is owing to the higher concentration of HO$_2$ due to reactions with reduced NO and less aerosol uptake (See Sec. 3.3.3).

3.2.2. Production and Destruction Rates of O$_x$



Figure S5 shows the spatial distribution of the changes in the mean daytime production rate of odd oxygen [$P(O_x)$] during January and July 2018 resulting from emission changes/reductions. In January, with the reduction of $NO_x$ emissions, the calculated decrease in $P(O_x)$ is the largest in southern and eastern China (by 1.5-2.0 ppbv h$^{-1}$ (15-20%); Fig. S5a). Some positive changes in the $O_x$ production rates are simulated in the metropolitan areas of the YRD and SCB, with

an increase of 0.8-1.5 ppbv h$^{-1}$(7-15%). These positive values are attributed to the increase in the concentration of $HO_2$ and $RO_2$ radicals, as $P(O_x)$ is strongly contributed by the reaction rates of NO with the $HO_2$ and $RO_2$ radicals. With the reduction in AVOCs emissions, $P(O_x)$ decreases by up to 4.5 ppbv h$^{-1}$ (42%) in the PRD region (Fig. S5c). When considering combined reductions of the $NO_x$ and AVOCs emissions, the calculated daytime value of $P(O_x)$

is reduced by 2-3 ppbv h$^{-1}$ (30-60%; Fig. S5e) in the whole of China. When the reduction in all anthropogenic emissions is considered, the ozone production rate decreases by 3-5 ppbv h$^{-1}$ (30-70%; Fig. S5g). In different monitoring sites, the highest reduced value of $P(O_x)$ is up to 4.2 ppbv h$^{-1}$ (40%) at the Guangzhou site, followed by 2.8 ppbv h$^{-1}$ (55%) in the Shanghai site, in the effect of AVOCs emission reduction only (Fig. S6).


The decrease of $P(O_x)$ for summer conditions is more significant than for winter conditions. With $NO_x$ emissions reduced by 50% (Fig. S5b), the reduction in the $O_x$ production rate is within the range of 15-20 ppbv h$^{-1}$ (20-50%). As shown in Fig. S6, the value of $P(O_x)$ decreases by 50% in Guangzhou, 45% in Shanghai, 30% in Chengdu, and 25% in Beijing. In response to

a 50% reduction in AVOCs emissions, the decrease in the value of $P(O_x)$ (5-10 ppbv h$^{-1}$ (10-28%); Fig. S5d) is smaller than the decrease resulting from the reduction in $NO_x$ emission. We also found a decrease of P(Ox) by 28% in Guangzhou, by 10% in Shanghai, by 25% in Chengdu, and by 20% in Beijing (Fig. S6). When the reduction in both $NO_x$ and AVOCs emissions is applied, the reduced value of $P(O_x)$ is around 16-25 ppbv h$^{-1}$ (22-55%; Fig. S5f).

With further reduction of other emissions, the values of $P(O_x)$ decrease by 15-22 ppbv h$^{-1}$ (20-52%; Fig. S5h).

Figure S7 shows the corresponding changes in the destruction rate of odd oxygen ($D(O_x)$). The decrease of $D(O_x)$ resulting from the 50% reduction of $NO_x$ is in the range of 0.1-0.5 ppbv h$^{-1}$

(20-30%) in winter (Fig. S7a), with the highest reduction simulated in the PRD regions. In summer, the reduction of $D(O_x)$ reaches 1.0-2.0 ppbv h$^{-1}$ (15-25%), with the largest decrease occurring in the NCP (Fig. S7b). The decrease of $D(O_x)$ due to the reduction in AVOCs emissions is most prominent in the urbanized areas, with the highest decrease reaching 0.2 ppbv h$^{-1}$ in winter (25%, Fig. S7c) and 0.8 ppbv h$^{-1}$ (10%) in summer (Fig. S7d).


The budgets of radicals and ozone are important indicators for the formation potential of ozone and other secondary pollutants (Tan et al., 2019). The decrease in the values of the summertime $P(RO_x)$ and $P(O_x)$ provides some information that can be useful to mitigate secondary photochemical pollution. The changes in the budgets due to emission reduction suggest that

the most effective way to reduce summertime $P(RO_x)$ and $P(O_x)$ is to reduce $NO_x$ emissions. When applying a combined reduction in the $NO_x$ and AVOCs emissions, the decrease of the $P(RO_x)$ and $P(O_x)$ values is further enhanced. Thus, reductions in specific AVOCs emissions are needed to conduct the effective mitigation of air pollution in China.



### 3.3. Changes in the concentrations of ozone and other secondary pollutants


Tropospheric ozone is a secondary pollutant, and its formation is largely affected by the levels of ozone precursors, including $NO_x$, VOCs, and CO (Wang et al., 2022). The reduction in the emissions of these primary species leads to changes in the photochemical formation of ozone. The formation of secondary inorganic aerosols (SIA), including particulate nitrate ($NO_3^-$), sulfate ($SO_4^{2-}$), and ammonium ($NH_4^+$), is associated with the level of their gas-phase precursors, such as $NO_2$, $SO_2$, and $NH_3$, and oxidants (Zheng et al., 2015). The oxidizing processes of AVOCs, including benzene, xylene, and toluene, play an important role in the formation of anthropogenic secondary organic aerosol (SOA) (Hu et al., 2017). Owing to high uncertainties in the chemical mechanisms adopted in chemical transport models, only limited studies have assessed the impact of reduced primary emissions on the tropospheric concentration of SOA (Hu et al., 2017; Li et al., 2022). In this section, we quantify the response of ozone precursors, ozone, and secondary aerosols to the reduction in the emissions of primary pollutants.


### 3.3.1. Precursors and intermediates in ozone formation

Firstly, we describe the changes in the surface concentration of ozone precursors and intermediates, including NO, $NO_2$, OH radical, $HO_2$ radical, and specific species of hydrocarbons and OVOCs in response to different reductions in surface emissions.


Figure 5 displays the spatial distribution of the responses of the ozone precursors and intermediate to the reduction of $NO_x$ emissions in the daytime. In January, a decrease of NO (by up to 6 ppbv (40%); Fig. 5a) and $NO_2$ (by up to 8 ppbv (25%); Fig. 5c) is derived in the metropolitan regions of China, which is consistent with the spatial distribution of reduction in $NO_x$ emissions (Fig. S1a). The calculated mixing ratio of surface OH radical decreases by 0.05 pptv (40%) in southern China (Fig. 5e), where the ozone sensitivity is controlled by $NO_x$ (Fig. 1c), is due to the decrease in the oxidation capacity of atmospheric (See Sec.3.4.2). At the same time, a relatively strong increase of OH is found in the PRD region (by 0.03 pptv; 24%), due to less consumption by the reduction in $NO_x$ and more production from the photolysis of enhanced OVOCs (Fig. 6g). An increase in the surface mixing ratio of the $HO_2$ radical is derived in southern China (by up to 5 pptv (60%); Fig. 5g), which is associated with a decrease in the aerosol load (see Sect. 3.3.3), and hence in reduced $HO_2$ uptake (Song et al., 2021).


In July, the decrease in the concentration of NO (4 ppbv (50%); Fig. 5b) and $NO_2$ (6 ppbv (35%); Fig. 5d) due to the reduction in the $NO_x$ emission is smaller than in winter, which is related with the smaller reduction in summertime $NO_x$ emissions (Fig. S1g). The level of summertime decreases in OH radicals (by up to 0.15 pptv (30-40%); Fig. 5f) is larger than the decrease derived in winter and displayed in broader areas. The spatially varied distribution in the changes of OH and $HO_2$ radicals in two seasons can be explained by the seasonal variations of UV, water vapor, and solar radiation, with high values concentrated in southern China only in winter and evenly distributed in the whole of China in summer (Dai et al., 2023). An increase





in the surface mixing ratio of $HO_2$ radical is derived in the NCP (6-8 pptv; 15-20%) and is due to the decreased uptake by the aerosol surfaces. An increase of alkene is calculated in July by 0.2-0.5 ppbv (10-20%), owing to the reduced consumption by decreased OH radicals.

For the 50% decrease in AVOCs emissions, the calculated surface concentration of alkenes decreased by up to 3.0-4.0 ppbv (30-40%; Fig. 7a) in urban China in January. At the same time, a decrease is calculated in the mixing ratio of OH (by 0.005-0.015 pptv (4-12%); Fig. 7c) and $HO_2$ radicals (by 3pptv (36%); Fig. 7e) in the southern part of China. Reasons for the decreases in these radicals are the declined contributions from the VOCs oxidation and OVOCs photolysis. Owing to the reduced consumption of hydrogenated radicals, an increase of NO (by 1.2-2.0 ppbv (7-12%)) is derived in the urban areas of China (Fig. S8a). In July, the calculated decrease of alkenes is estimated to be 1.0-2.0 ppbv (30-50%; Fig. 7b). The reduction in VOCs also leads to a summertime decrease in radicals in urban areas, with the decrease of OH by 0.03-0.05 pptv (8-12%) and $HO_2$ radicals by 3-5 pptv (6-10%). The relevant summertime enhancement in NO concentration is 0.8-1.0 ppbv (10-15%; Fig. S8b).

When applying the combined 50% reduction in $NO_x$ and AVOCs emissions, the changes in the spatial distribution of radicals are similar to the changes derived when a reduction was applied only to the $NO_x$ emissions alone. Compared with the results in the reduction $NO_x$ emissions alone, a smaller increase of OH radical is provided in urban China in January (by 0.01 pptv (12%); Fig. 8a) and in July (by 0.07 pptv (15-20%); Fig. 8b), which is attributable to the less production of OH radical from the VOCs oxidation due to reduction in AVOCs emission. The decrease of OH radical is also suppressed to 0.02 pptv (16%) in January and 0.1 pptv (10-12%) in July, which is due to the lower destruction rate of OH from the reaction with reduced $HO_2$ radicals. As the negative effects of declined VOCs oxidation due to AVOCs emissions reduction, the enhanced $HO_2$ radicals in the $NO_x$ emission reduction case are largely offset. As shown in Fig. 8c and d, the combined case derived less than 1 pptv (12%) of wintertime enhancement in the southern coastal areas of China and 3 pptv (6%) of summertime increase in the urban areas.

When applying a 50% reduction to $NO_x$, AVOCs, and other anthropogenic emissions, the mixing ratio of OH radical is positively varied compared with the results in the combined case (N+A case). A distinct increase of wintertime OH radical is derived in the PRD and SCB regions (by up to 0.03 pptv; Fig.8e). This increase is owing to the lower consumption of OH radical by the reduced concentration of $NH_3$, $SO_2$, and CO, due to their reduced emissions. In July, the increase of OH radical is also enhanced to 0.01 pptv (Fig. 8f) in VOC-limited areas. At the same time, the decreases of OH radicals in $NO_x$-limited areas are suppressed in both winter and summer, indicating the increasing atmospheric oxidative capacity in these areas. For $HO_2$ radicals, the additional emissions reduction also contributed to an increase in its mixing ratio, with a pronounced increase in southern China in January (by 1.5 pptv (18%) Fig 8g) and in the NCP region in July (by 6 pptv (12%); Fig. 8h). The enhancement in HO2 radicals is caused by less $HO_2$ loss via the aerosol uptake, as a decrease is derived in aerosol concentration due to the reduction in the precursor of secondary inorganic aerosols (See Sec. 3.3.3).





OVOCs are both contributed by the primary source, their direct emissions, and the secondary source, the oxidation of hydrocarbons (Wang W. et al., 2022). With the reduction in emissions, the concentrations of OVOCs also change substantially. Figure 6 g and h displays the spatial distribution of the changes in total OVOCs due to the 50% reduction in $NO_x$ emissions in January and July of 2018. During January, there is an increase in the calculated concentration of OVOCs in the NCP, YRD, and PRD. In these areas, the highest wintertime increases in the concentration of OVOCs reach about 0.5-1.0 ppbv (5-10%; Fig. 6g). At specific sites, the highest increase in OVOCs concentrations is calculated at the Shanghai and Guangzhou sites. As shown in Fig. 9, at these two urban sites, the reduction in $NO_x$ emissions leads to an increase in OVOCs concentration by 1.8 ppbv (12%) and 1.2 ppbv (8%), respectively, which is mainly contributed by the increase of HCHO (Fig. 6e) due to the higher secondary formation from the oxidation of VOCs (Li et al., 2021). When AVOCs emissions are reduced, a decrease in OVOCs concentration is simulated in most regions of East China (by up to 8 ppbv (12%); Fig. 7g), especially the SCB and PRD regions, with the highest decrease in ketones (by 2-3 ppb (10-15%); Fig. 9).

In summer, with the 50% $NO_x$ emission reduction, the increases of simulated OVOCs concentrations in North China are more pronounced than in the wintertime, with an increase of 0.5-1.5 ppbv (10-15%; Fig. 6h). These OVOCs increases in $NO_x$-limited areas are associated with the enhanced hydrocarbons concentration, including alkenes (Fig. 6d) and isoprene (Fig. S10a) due to less loss via the oxidation by OH radicals. While, in VOC-limited areas, the increase of OVOCs can be attributed to the increased mixing ratio of OH radical (Fig. 5f) due to less titration effect. With the 50% AVOCs emission reduction, similarly with the wintertime change, the simulated OVOCs decrease reaches up to about 5 ppbv (20%; Fig. 7h), with the largest contribution calculated in ketones (by 1~3 ppbv (8-15%); Fig. 9). Limited changes (less than 5%) are found in the concentration of OVOCs in the sub-rural sites Heshan, which is relevant to the major contribution of BVOCs, which are not notably influenced by anthropogenic emissions, to the secondary OVOCs formation at this site (Dai et al., 2023).

Quantification of the impact of emissions reduction on OVOCs abundance provides an important indicator of its impacts on the atmospheric oxidative capacity since the photolysis of OVOCs produces $HO_x$ radicals that drive fast ozone production and accelerate further VOC oxidation (Li et al., 2021). Li et al., (2021) reported that the fast photochemical production of ozone during the Covid-19 lockdown was attributed to the sharp reduction of the $NO_x$ emissions in urban China and the high VOCs emissions, driving ozone production through HCHO photolysis. The study also suggests extending the VOC emission controls year-round to avoid the spread of ozone pollution outside the summer season. In our study, with the 50% reduction of $NO_x$ emissions, the increased concentration of OVOCs, including HCHO, is simulated in both winter and summer, which is supportive of Li et al., (2021). The discussion in the following sections explores the changes in other secondary pollutants, including ozone and secondary aerosols.

3.3.2. Ozone



Figure 10 shows the changes in surface daytime ozone concentrations resulting from the 50% emissions reduction for January and July conditions. In winter, the reduction in $NO_x$ emission leads to an increase in the surface ozone concentrations, which is the largest in the YRD and PRD regions (8-10 ppbv (15-20%); Fig. 10a). During wintertime, as shown in Fig. 1a, a large

part of China is under a VOC-sensitive regime. In this $NO_x$-reduced case, the weakened titration effect due to the decrease in $NO_x$ concentrations favors ozone formation. In AVOCs emissions reduction case, a reduction in the surface ozone concentration, ranging from 2.0 to 8.0 ppbv (4-10%; Fig. 10c), is calculated in the southern part of China. This ozone decrease is owing to the decelerated ozone production rate attributed to the decrease in the concentration

of hydrocarbons (Fig. 7a) (Jacob et al., 1995) and lower $HO_x$ radicals originating from the photolysis of reduced OVOCs concentration (Fig. 7g).

In the combined emission reduction case, the wintertime ozone changes in a large part of China have primarily followed the ozone changes with $NO_x$ emissions reduction, with an ozone

increase of 3.0-7.5 ppbv (4-9%; Fig. 10e) in VOC-limited areas. As the distribution of ozone response mainly depends on the sensitive areas, in this combined case, a large part of China, including North China and some urban regions in South China, is located in VOC-limited areas (Fig. 1g). Thus, in these areas, the ozone response mainly follows the positive changes in the $NO_x$-reduction case. Simultaneously, a slight ozone decrease (by 2.0-4.5 ppbv; 5-8%) is

derived by the model over the southern coast of China. In these areas, the ozone sensitivity is under the control of the $NO_x$. Therefore, the ozone decrease is relevant to the negative ozone response in AVOCs emissions reduction case. Compared with the combined case, with further emission reduction in other species, including CO, $SO_2$, and $NH_3$, a smaller ozone increase (by 3-5 ppbv; 4-6%) is calculated in the southern part of China (Fig. 10g). One of the reasons for

the higher ozone formation is the reduced CO concentration, as an ozone precursor, due to the CO emission reduction. Besides, a higher level of $HO_2$, resulting from less aerosol uptake due to emission reduction of $NH_3$ and $SO_2$ (See Sec. 3.3.3), can also lead to higher ozone formation.

In summer, an increase in the surface ozone concentration by up to 12 ppbv (10~20%) is only

calculated in the urbanized NCP, YRD, and PRD (Fig. 10b), under the effect of $NO_x$ emissions reduction. These areas are typically located in VOC-limited conditions (Fig. 1b); thus, ozone increase can be well explained by reduced ozone titration. At the same time, in $NO_x$-limited areas, the calculated surface ozone concentration is reduced by 2 to 8 ppbv (3-10%). This ozone decrease is associated with less photochemical formation from reduced NO (Fig. 5b) and $NO_2$

(Fig. 5d) concentrations. With the reduction of AVOCs emissions, the surface ozone concentration decreases in whole areas of China by up to 8.0-12.0 ppbv (8-20%; Fig. 10d). Unlike wintertime ozone decrease, which primarily occurs in southern China, the summertime ozone decline shifts from the southern regions to the northern ones, including the NCP, the PRD, and the YRD regions. This spatial variation is consistent with the distribution of model-

derived $HO_2$ radicals' changes (Fig. 7f), indicating the importance of AVOCs to the formation of $HO_2$ radicals.

In the combined 50% reduction case of the $NO_x$ and AVOCs emissions, the surface ozone concentration decreases by up to 12 ppbv (15%; Fig. 10f) in $NO_x$-sensitive areas due to the



influence of the reduction in the NO$_x$ and AVOCs emissions. In VOC-sensitive areas, the increase of ozone associated with the positive impact of the reduction in NO$_x$ emissions is largely compensated by the negative effect of the reduction in AVOCs emissions. An ozone decrease is derived in almost all areas of China and ranges from about 5 to 12 ppbv (6-15%). One exception is found at the Guangzhou site, where ozone slightly increases by 0.5 ppbv (Fig.

S11). The important contribution of BVOCs to the formation of ozone in the PRD region has been documented in many studies (Zhang et al., 2023). One reason can be the role of BVOCs species, such as isoprene, becomes more important to ozone formation in this area, when NO$_x$ and AVOCs emissions are reduced (see Sec. 3.4.1).

When emission reduction is conducted to all species, the largest ozone decrease is slightly rebounded to 10 ppbv (15%) in NO$_x$-limited conditions and 11 ppbv (13%) in VOCs-limited conditions (Fig. 10h).

### 3.3.3. Aerosols


Figure 11 shows the changes in the daytime average concentrations of secondary inorganic aerosol (SIA; the sum of NO$_3^-$, NH$_4^+$, and SO$_4^{2-}$) due to emissions reduction in January and July. In January, the 50% reduction of NO$_x$ leads to a large wintertime decrease of SIA by up to 10-18 µg m$^{-3}$ (12-20%; Fig. 10a), with a dominant contribution from the decrease in NO$_3^-$

(by 5-12 µg m$^{-3}$; Fig. S12a) due to reduced concentration of NO$_2$ (Fig. 5c), followed by NH$_4^+$ (by 2-4 µg m$^{-3}$; Fig. S13a) due to reduced NO$_3^-$ concentration (Meng et al., 2022). In July, the decrease of SIA due to NO$_x$ emission reduction is smaller than during wintertime, with a decrease of 2-3 µg m$^{-3}$ (6-10%; Fig. 11b). With a 50% reduction of AVOCs emissions, the simulated reduction in concentration of SIA is limited, with the decrease less than 4 µg m$^{-3}$

(4%; Fig. 11c) and 1.5 µg m$^{-3}$ (5%; Fig. 11d) in winter and summer, respectively. With a combined 50% emission reduction in NO$_x$ and AVOCs in winter, as shown in Fig. 11e, the decrease in SIA is larger than the decrease affected by separated emission reduction. This enhanced decrease is also derived from the decrease of NO$_3^-$ (Fig. S12e), NH$_4^+$ (Fig. S13e), and SO$_4^{2-}$( Fig. S14e) due to combined emission reduction. In specific sites (Fig. 13), the

strengthened aerosol decrease can be found in six sites, except for the Guangzhou and Heshan sites, where BVOCs play an important role in oxidative capacity. This phenomenon indicates that the reduction in AVOCs emission will increase the efficiency of aerosol decrease in winter when NO$_x$ emission is reduced.

With the further reduction in other emissions, including the emissions of NH$_3$ and SO$_2$, the decrease in the SIA is largely enhanced. In January, the decrease of the SIA concentration reaches 15-25 µg m$^{-3}$ (20-38%; Fig. 11g), while in July, it reaches 8-10 µg m$^{-3}$ (30-40%; Fig. 11h). This enhancement in the aerosol decrease is caused to a large extent by the decrease in the concentration of NH$_4^+$ (Fig. S13 g and h) and SO$_4^{2-}$ (Fig. S14 g and h).


Figure 12 shows the changes in the daytime average concentrations of secondary organic aerosol (SOA) due to emissions reduction in January and July. With 50% NOx emission reduction in winter, a slight increase of SOA (1.0-1.5 µg m$^{-3}$ (3-5%; Fig. 12a)) is derived in the



urban areas of NCP, YRD, and PRD regions, which is consistent with the increase in oxidants,
including ozone and OH radicals. When AVOC emissions are reduced, the simulated decrease
in SOA concentration is about 2-5 μg m$^{-3}$ (5-11%; Fig. 12c) in the southern part of China. In
summer, the decreased level of SOA is comparable to the wintertime decrease. However, the
simulated reductions in SOA undergo a spatial shift, from the southern part of China in winter
to the northern China Plain. Reasons for the seasonal variations in spatial pattern are the spatial
distribution of SOA precursors, including oxidants, hydrocarbons, and OVOCs, and the
seasonally changing meteorological parameters, such as temperature and solar radiation.

The aerosol effect on ozone formation has been discussed in several modeling studies (Li et
al., 2019; Liu et al., 2020; Dai et al., 2023). Our results show that the SIA concentration is
largely decreased in winter due to $NO_x$ emissions reduction, while the declined SOA
concentration relies more on the reduction of AVOCs emissions. This aerosol decrease
weakens the aerosol extinction effect and therefore enhances the ozone photochemical
formation rate (Tan et al., 2022). At the same time, aerosol decreases can result in a decline in
aerosol uptake through heterogeneous reactions, resulting in an enhanced concentration of $NO_2$
and $HO_2$ radicals. Moreover, the formation of HONO will also decline with $NO_x$ emission
reduction (Fig. 6 a and b) through the heterogeneous conversion from $NO_2$, photolysis of $NO_3^-$
and direct emission from transport (Dai et al., 2023). Thus, the atmospheric level of the OH
radical will also be reduced, as the photolysis of HONO is an important source of $HO_x$ radicals.
However, the effect of the increase of $NO_2$ and the decrease of OH radical associated with the
decrease of aerosol, is negligible, due to the offset effect by the decrease in the concentration
of $NO_2$ when $NO_x$ emissions are reduced. The enhancement in $HO_2$ radicals, contributed by
aerosol decrease, is consistent with the weakened titration effect due to reduced $NO_2$
concentration by $NO_x$ emission reduction and can favor the formation of secondary pollutants,
including $O_3$ and SOA, in the VOC-limited areas. When combined with a decrease in AVOCs
emissions, the increase of secondary pollutants, due to reduced $NO_x$, can be largely offset by
the decrease in hydrocarbons and OVOCs. Exceptions can be found in the areas where high
enough UV and temperature for the photochemical oxidation processes of VOCs are lacking,
such as the northern part of China during wintertime, and where the role of BVOCs is crucial
for the formation of ozone and secondary aerosols, including the PRD region during
wintertime. A schematic diagram describing the chemical mechanisms involved in the
atmospheric ozone chemical in response to the reduction in $NO_x$ emissions is shown in Fig. 14.

3.4. Changes in the photochemical reactivity and the atmospheric oxidative capacity

The changes in photochemical parameters, including the OH reactivity and the atmospheric
oxidative capacity (*AOC*), in response to the adopted reductions in emissions are discussed in
this section. The OH reactivity (*$OH^R$*) is expressed as the OH loss frequency due to the reactions
by VOC (*$VOC^R$*) (including CO) and $NO_x$ (*$NO_x^R$*). Thus, this parameter can be used to represent
the specific role of VOCs, $NO_x$, as well as CO that determine the photochemical formation of
ground-level ozone (Tan et al., 2017, Xue et al., 2016; Wang et al., 2022). Atmospheric
Oxidizing Capacity (*AOC*) is a parameter that characterizes the self-cleansing ability of the
atmosphere (Liu et al., 2022) and is derived here as the rate at which CO, methane ($CH_4$), and



non-methane hydrocarbons (NMHCs) are oxidized by atmospheric oxidants, including OH, $O_3$, and $NO_3$ (Xue et al., 2016; Dai et al., 2023). This parameter allows us to characterize the removal process of secondary species, including $O_3$, SIA, and SOA, and therefore, can be used as an indicator to design control policies for secondary pollutants. A detailed description of these parameters can be found in Part 1 of the paper (Dai et al., 2023).

### 3.4.1. OH reactivity

Figure 15 displays the daytime changes in OH reactivity for VOCs ($VOC^R$) in January and July in response to the specified emission reductions. During wintertime, not surprisingly, the response of $VOC^R$ to the $NO_x$ reduction is considerably smaller than the response to the reduction of AVOCs emissions. Reduction in the $NO_x$ emission slightly reduces the value of $VOC^R$ by 0.2-0.5 $s^{-1}$ (<5%) in the YRD and PRD regions (Fig. 15a). The reduction in $VOC^R$ resulting from the 50% reduction in the AVOCs emissions is found to be of the order of 1.5-3 $s^{-1}$ (12-25%; Fig. 15c). In summer, the reduction in the $NO_x$ emission increase $VOC^R$ in the southern and northeastern parts of China (1.5-2 $s^{-1}$ (8-15%); Fig. 15b), which is consistent with the increase in the concentration of reactive VOCs species including alkenes (Fig. 6d) and isoprene (Fig. S11a) due to reduced oxidant. The potentially positive role of isoprene in the $VOC^R$ in this case highlights the increasing importance of BVOCs in the summertime atmospheric oxidative capacity as the anthropogenic emissions are reduced. For reduced AVOCs emissions, a decrease in $VOC^R$ of 2-3 $s^{-1}$ (10-20%) is derived in the four major metropolitan regions of China, i.e. the NCP, YRD, PRD, and SCB (Fig. 15d).

Figure S15 shows the changes in the value of $NO_x^R$ in response to the 50% reductions in NOx emissions for the January and July conditions. The reduction in $NO_x$ emissions leads to a decrease in $NO_x^R$ in the range of 4.0-6.0 $s^{-1}$ (30-50%) during January (Fig. S15a) and of 3.0-4.0 $s^{-1}$ (35-45%) during July (Fig. S15b). The higher decrease in winter is consistent with the larger wintertime decrease in NO and $NO_2$. When considering the reduction in AVOCs emissions, the changes in $NO_x^R$ are relatively small (less than 3% in both January and July). When combining the emission reductions, the changes in the OH reactivity are substantially larger than individual impacts. For example, the combined reduction leads to a wintertime decrease in $NO_x^R$ up to 8 $s^{-1}$ (65%; Fig. S15e). For the 50% reduction in total anthropogenic emissions, the OH reactivity further decreases in urban areas, which is attributed to the decrease in the concentration of CO due to the relevant emission reduction (Fig. S16d).

For different sites, in winter, the largest decrease is found to occur at the Beijing site (Fig. 16a), with a decrease of 50% in OH reactivity in the combined 50% reductions in VOCs and $NO_x$ emissions. In the combined case, the reduction in the OH reactivity is 45% in Shanghai, 40% in Chengdu, 35% in Wangdu, and 30% in Guangzhou. In summer, the highest decrease also occurs at the Beijing site (by 45%) when a 50% reduction is applied to all emissions. Under these conditions, the OH reactivity is reduced by about 30% in Guangzhou, 25% in Chengdu, and 15% in Shanghai. Besides, the contribution of BVOCs to $OH^R$ becomes considerably larger, with the reduction in $NO_x$ emissions. The most significant increase is calculated at the Heshan sites, with the contribution increasing from 18% to 34%. At the Beijing and Guangzhou



sites, the contribution also increases from 3-5% to 10-15%. Small reductions of $OH^R$ (less than 3%) are found at background sites (Waliguan), which suggests that the impact of anthropogenic emissions on the background OH reactivity is limited.

705

3.4.2. Atmospheric oxidative capacity

Finally, we show the changes in the spatial distribution of daytime *AOC* resulting from the adopted 50% reduction in the emissions for the January and July conditions. In January, the reduction in $NO_x$ emission leads to an increase in *AOC* (by $0.6 \times 10^7$ molecules cm$^{-3}$ s$^{-1}$ (18%); Fig. 17a) in the metropolitan areas of the YRD, PRD, and SCB and a decrease in *AOC* in southern China (by $0.9 \times 10^7$ molecules cm$^{-3}$ s$^{-1}$; 25%). The increase in the daytime *AOC* of the metropolitan areas is consistent with the increase in the concentration of surface $O_3$ (Fig. 10a) and SOA (Fig. 12a), which is related to the increase in the concentration of oxidants, including OH and $HO_2$ radicals, ozone, and OVOCs. With the reduction in the AVOCs emissions, the daytime *AOC* is reduced in all the major regions of China (Fig. 17c). The large decreases occurred in the southern part of China (by $1.0 \times 10^7$ molecules cm$^{-3}$ s$^{-1}$; 30%), which is consistent with the large decrease of $HO_x$ radicals and ozone due to reduced hydrocarbons and OVOCs. With a 50% emission reduction in $NO_x$, AVOCs, and other primary pollutants (Fig. 17g), the level of daytime *AOC* is higher than the level with the decrease in $NO_x$ and AVOCs emissions only (Fig. 17e). The higher level of daytime *AOC* is caused by the enhanced OH and $HO_2$ radicals (Fig. S9 e and g) resulting from less consumption by the reduced CO, $SO_2$, and aerosol uptake.

When examining the wintertime impact of $NO_x$ emission reduction at different sites, the largest daytime *AOC* increase occurs at the Shanghai site (by 20%) (Fig. 18b), followed by the sites of Guangzhou (by 18%), Chengdu (by 12%), and Beijing (by 8%). These results are consistent with the wintertime increase of OH radical (Fig. 5e). With the reduction in AVOCs emissions, the largest *AOC* decrease is found at the Guangzhou site (by 50%), followed by the Shanghai (by 48%) and Beijing sites (by 40%). The decrease in *AOC* is mainly attributable to the reduced contribution from the reactions between OH with alkenes, followed by OH with aromatics and OVOCs.

During summertime, the decrease in daytime *AOC*, resulting from the reduction in emissions, is more pronounced than in wintertime. With a 50% reduction in $NO_x$ emissions, a decrease in *AOC* is derived in large areas of China ($NO_x$-limited regimes), with a peak decrease of $1.5 \times 10^7$ molecular cm$^{-3}$ s$^{-1}$ (20%; Fig. 17b) derived in the NCP region. At the same time, an increase of daytime *AOC* by $2.5 \times 10^7$ molecules cm$^{-3}$ s$^{-1}$ (5%) is predicted in urban areas (VOC-limited regimes). The spatial pattern of predicted changes in *AOC* due to reduced $NO_2$ is consistent with the changes in OH (Fig. 5f), HCHO (Fig. 6f), and $O_3$(Fig. 10b), indicating the efficient representation of *AOC* in the formation of secondary pollutants. At specific sites, the simulated increase (by 5-7%) is derived in the sites of Guangzhou (Fig. 18c), Shanghai, and Chengdu. The decrease of daytime *AOC* due to the reduced AVOCs emissions is most pronounced in urbanized areas with values reaching $2 \times 10^7$ molecules cm$^{-3}$ s$^{-1}$ (25%; Fig. 17d), in the NCP, YRD, and PRD. At four urban sites (Fig. 18), the decrease in *AOC* ranges from 15% to 30%.



When combining the 50% reduction in AVOCs and $NO_x$ emissions, the decrease in *AOC* is derived in the whole of geographical China (Fig. 17f). The decreases in different sites range from 10%-20% at the four city sites, and 16% and 20% at the rural sites; a reduction of around 5% is derived at the two remote sites. When adding the decrease in other emissions, the spatial
distribution of daytime *AOC* decrease is not substantially different from the previous cases.

For nighttime (20:00 to 05:00 LST) *AOC*, the reduction in $NO_x$ emissions results in an increase of *AOC* by up to $2\times10^6$ molecules $cm^{-3}$ $s^{-1}$ in both January (50%; Fig. 19a) and July (10%; Fig. 19b) due to the reaction of alkenes with $O_3$. The increases are mainly located in urban China
in winter and in southern and northern China in summer, which is associated with the relevant enhancement in the concentration of ozone (Fig. 10 a and b) and summertime alkenes (Fig. 6 d). Moreover, the most significant increase from the contribution of alkene ozonolysis is also derived in city sites. For example, in January, the contribution of ozonolysis to nighttime *AOC* increases from 31% to 40% in the Shanghai site (Fig. 20b; Fig. S17b) and from 10% to 16% in
the Chengdu sites (Fig. 20d; Fig. S17d). In summer, the changes are smaller than in winter, with the increase from 10% to 14% in the Beijing site (Fig. 20a; Fig. S17a).

Notably, as shown in Fig. 18, among all the emission reduction scenarios, the largest decrease in daytime *AOC* at urban sites is provided with a 50% reduction in AVOCs emission. To
mitigate the formation of secondary pollution, the reduction in AVOCs emissions needs to be implemented. In practice, a reduction of 50% AVOCs emissions is hard to achieve in the short term, thus efficient control over emissions of key AVOCs species is needed based on their contributions to *AOC*. As shown in Fig. S18, at four metropolitan sites (Beijing, Shanghai, Guangzhou, and Chengdu), the largest contribution to daytime *AOC* is provided by the reaction
of OH with alkenes (30-44%), carbon monoxide (28-40%) and aromatics (15-25%) in winter. In summer, a higher contribution from OH with OVOCs (30-40%) is derived at four urban sites, in comparison to the wintertime conditions (12-22%). The contributions from OH with alkenes and aromatics are around 15-30% and 10-20%, respectively. Combined with the consideration of the dominant contribution of alkenes and aromatics to OH reactivity, we
suggest that reducing the emissions of alkenes, aromatics, and unsaturated OVOCs should be prioritized in the control of *AOC* and secondary pollutants in urban China. To give more accurate suggestions for AVOCs emission control, more studies on the contribution of specific AVOCs emissions to *AOC* are required.

**4. Summary**

Our analyses of the model results provide insight into the changes in the atmospheric oxidizing capacity and chemistry in China in response to reductions of anthropogenic emissions.

With a 50% reduction in $NO_x$ emissions, the production rate of radicals and odd oxygen decreases substantially in both January and July. These decreases are larger in summer (by 0.5-1.2 ppbv $h^{-1}$ (12-28%) and 15-20 ppbv $h^{-1}$ (20-50%)) than in winter (by 0.3-0.7 ppbv $h^{-1}$ (30-33%) and 1.5-2.0 (15-20%) ppbv $h^{-1}$) and can be attributed to the reduction in the concentrations of nitrogen oxides and OH radicals. During wintertime, however, an increase





by 0.8-1.5 ppbv h$^{-1}$ (7-15%) is derived to produce odd oxygen in the PRD region, which is related to the increasing concentration of the HO$_2$ radical resulting from the increase in the OVOCs concentration and the reduction in the aerosol load. Moreover, during summertime, the simulated concentration of surface ozone decreases in NO$_x$-limited areas (up to 2-8 ppbv (3-12%)) and increases in VOC-limited areas (up to 12 ppbv (15%)). The increasing ozone

concentration in VOC-limited areas is attributed to the reduced titration by NO$_x$ decrease. In the NO$_x$-limited areas, the decrease in the ozone concentration results from the reduced photochemical ozone formation associated with the reduction in the concentration of NO$_x$ with considerate effects from reduced HO$_x$ radicals, due to less formation from the photolysis of HONO and OVOCs.


With the AVOCs emissions lowered by 50%, a lower concentration of alkenes by up to 4.0 ppbv (40%) in January and by 2.0 ppbv (50%) in July is derived in urban China areas. The concentrations of OVOCs are reduced by 2.0-5.0 ppbv (8-20%) in January and by 3.0-5.0 ppbv (12-20%) in July due to the reduced primary emissions and secondary formations from the

oxidation of hydrocarbons. Results show that, with a 50% reduction in AVOCs emissions, the RO$_x$ and O$_x$ production rates also reduce; the decreases reach 0.1-0.8 ppbv h$^{-1}$ (10-35%) and 1.0-4.5 ppbv h$^{-1}$ (20-42%), respectively, during winter, and 0.3-0.8 ppbv h$^{-1}$ (7-18%) and 5.0-10.0 ppbv h$^{-1}$ (10-28%), respectively, during summer. It is attributed to a decrease in the concentration of HO$_2$ radicals, associated with a decrease in the oxidation of hydrocarbons and

photolysis of OVOCs. The simulated ozone concentrations are reduced by 2.0-8.0 ppbv (4-15%) in southern China during winter and by 8.0-12.0 ppbv (9-20%) in urbanized areas during summer.

With the combined 50% reduction in NO$_x$ and AVOCs emissions, the geographic patterns of

the simulated ozone changes in January resemble more the patterns associated with NO$_x$ emission reduction than those with AVOCs emission reduction. Generally, the surface ozone concentration increases by 5-12 ppbv (6-15%) in urbanized areas. In summer, the changes in the ozone concentrations in VOC-limited areas are affected positively by the reduction in NO$_x$ emissions and negatively by the reduction in AVOCs emissions. The net effect is an ozone

reduction in almost all areas of China except the PRD region, where the role of biogenic VOCs emissions is considered to be comparable to or more important than anthropogenic emissions.

For aerosol response to primary emissions reduction, a large decrease (up to 20%) in SIA concentrations is derived with a 50% NO$_x$ emissions reduction in winter, dominantly

contributed by decreases in NO$_3^-$, followed by NH$_4^+$. With a 50% reduction in AVOCs emissions, the simulated SIA decrease is limited (less than 5%) during wintertime. However, when combining the emissions reduction in NO$_x$ with AVOCs, a strengthened effect can be found in the decrease of each composition of SIA (NO$_3^-$, NH$_4^+$, and SO$_4^{2-}$), indicating the necessity of a joint reduction in NO$_x$ and AVOCs emissions in an efficient reduction in the

aerosol concentration in the future. The decrease of SOA is more sensitive to AVOCs emissions reduction, as the oxidation of several AVOCs is an important source of SOA formation. A geographical shift can be found in the spatial distribution of SOA decrease, with large decreases in southern China during wintertime and in the NCP region during



summertime. This shift is consistent with the seasonally varied distribution of $HO_x$ radicals. With the aerosol decrease, the effect of the associated enhancement of $HO_2$ will be consistent with the weakened nitration effect by $NO_x$ emission reduction and lead to an enhancement in secondary pollutants. When combined with the reduction in AVOCs, the positive effect of aerosol-enhanced $HO_2$ will still be important to ozone formation in northern China during wintertime and PRD regions during summertime, due to the relatively small influence of AVOCs.

Regarding the OH reactivity from VOCs in urban areas, we find that a 50% decrease in the $NO_x$ emission leads to a decrease of 0.2-0.5 $s^{-1}$ (<5%) and 0.6-3.0 $s^{-1}$ (5-20%) in January and July, respectively. During summertime, an increase of 3.0 $s^{-1}$ of VOC-related OH reactivity by up to 3.0 $s^{-1}$ s is found in southern and northeastern China. This increase is associated with an increase in the concentrations of alkenes and biogenic VOCs (e.g. isoprene), due to less consumption of these reactive VOCs by the reduced OH radical. Regarding the OH reactivity from $NO_x$, a slight increase is derived during wintertime in the specific case where AVOCs emissions are reduced by 50%. This increase is attributed to the increase in the calculated $NO_2$ concentrations.

For atmospheric oxidative capacity (*AOC*), a reduction in $NO_x$ emissions during January leads to a slight decrease (0.2-0.4 $\times 10^7$ molecules $cm^{-3}$ $s^{-1}$; 3-6%) in the daytime *AOC* value of southern China and a distinct increase (1.0$\times 10^7$ molecules $cm^{-3}$ $s^{-1}$; 18%) in the PRD region, due to the enhanced $HO_x$ radical from the increase alkenes ozonolysis and OVOCs photolysis. In July, *AOC* decreases by 0.4-1.5 $\times 10^7$ molecules $cm^{-3}$ $s^{-1}$ (7-20%) in the $NO_x$-limited areas and increases by 2.0$\times 10^7$ molecules $cm^{-3}$ $s^{-1}$ (25%) in the VOC-limited areas when $NO_x$ emissions are reduced by 50%. These changes are linked to the corresponding changes in the levels of OH radicals and ozone. A reduction in the AVOCs emissions leads to a reduction in the daytime *AOC* by 1.0$\times 10^7$ molecules $cm^{-3}$ $s^{-1}$ (13%) in January and 2.0$\times 10^7$ molecules $cm^{-3}$ $s^{-1}$ (25%) in July. Specifically, the summertime decreases result predominantly from the weakening of the reaction between OH and alkenes, followed by the reaction of OH with aromatics and OVOCs.

Our results suggest that a substantial reduction in the $NO_x$ emissions and associated effects in the aerosols decrease, the photolysis of OVOCs and ozonolysis of alkenes will lead to an enhancement in surface ozone concentration, OH reactivity from VOCs and *AOC* parameters in urbanized areas. A coordinated strategy for the reduction in $NO_x$ and AVOCs emissions is required to efficiently reduce the ozone levels in metropolitan areas. More detailed investigations to characterize the contribution of individual VOCs to summertime ozone formation are required to develop efficient mitigation strategies against rising ozone concentrations and the related change in the oxidative capacity of the atmosphere in China.





*Code and data availability*. The WRF-Chem model is publicly available at https://www2.mmm.ucar.edu/wrf/users/. The air quality data at surface stations are publicly available at the website of the Ministry of Ecology and Environment of the People's Republic of China at http://english.mee.gov.cn/.


*Author contributions.* JD and GB designed the structure of the manuscript, performed the numerical experiments, analyzed the results, and wrote the manuscript. JD analyzed the data and established the figures. All co-authors provided comments and reviewed the manuscript.

*Competing interests.* At least one of the (co-)authors is a member of the editorial board of Atmospheric Chemistry and Physics.

*Acknowledgments*. The present joint Sino-German study was supported by the German Research Foundation (Deutsche Forschungs Gemeinschaft DFG), the National Science
Foundation of China (NSFC) under Air-Changes grant no. 4487-20203, the Research Grants Council– University Grants Committee (grant no. T24-504/17-N) and the NSFC (grant no.42293322). The National Center for Atmospheric Research (NCAR) is sponsored by the US National Science Foundation. We would like to acknowledge the high-performance computing support from NCAR Cheyenne.







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






Table 1. Sensitivity experiments

| Model Experiments | Description |
|---|---|
| *BASE* | Without emission reduction |
| *NOx* | With emission reduction in $NO_x$[a] by a factor of 2 |
| *AVOCs* | With emission reduction in anthropogenic VOCs[a] by a factor of 2 |
| *N+A* | With emission reduction in $NO_x$[a] and anthropogenic VOCs[a] by a factor of 2 |
| *TOTAL* | With emissions reduction in all species[a] by a factor of 2 |


[a] Relevant species is shown in Table S1.



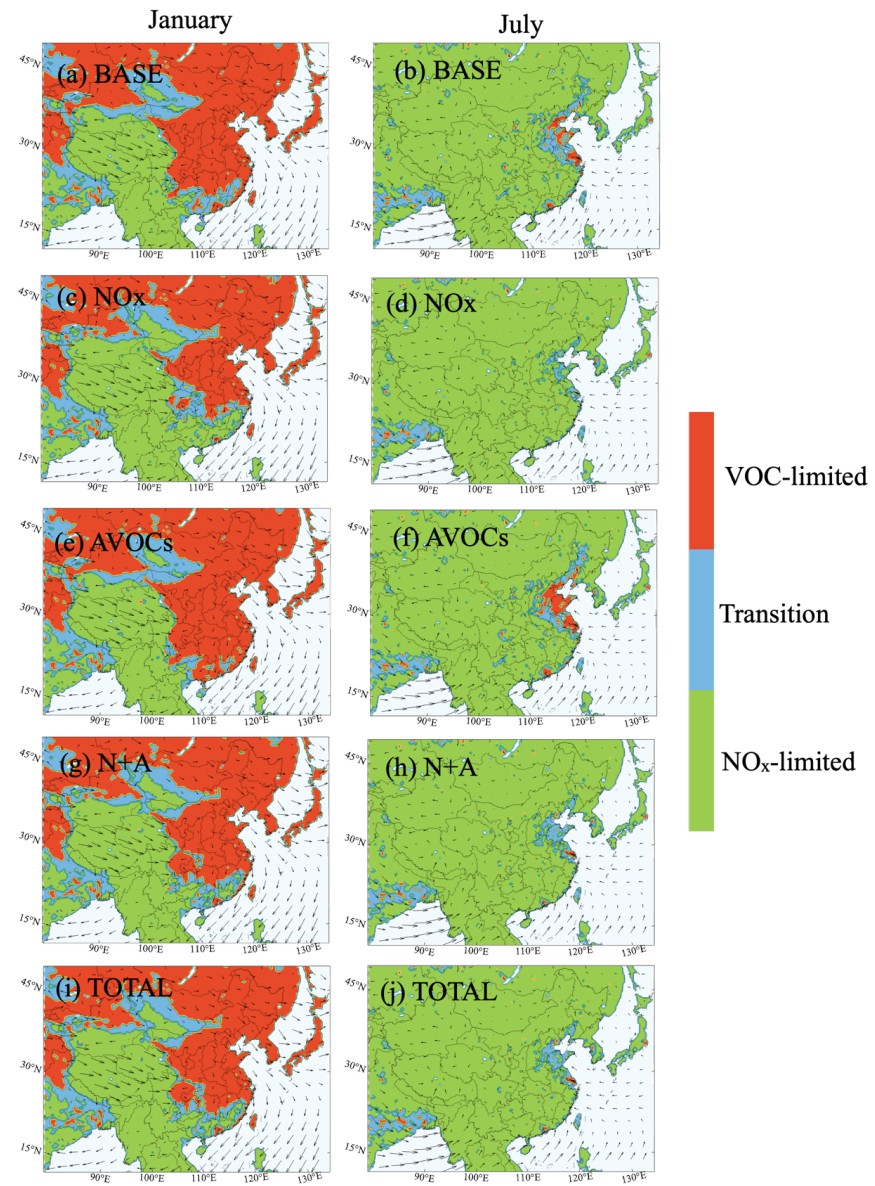

Figure 1. Display of regions in which ozone production is limited by the availability of nitrogen oxides (NO$_x$-limited, in green), and volatile organic components (VOC-limited, in red) from the BASE case (a, b), the NOx case (c, d), the AVOCs case (e, f), the N+A case (g, h), and the TOTAL case (i, j) in January (a, c, e, g, i) and July (b, d, f, h, j) of 2018. The regions where ozone production is controlled by the availability of both NO$_x$ and VOCs (transition) are shown in blue. The indicator used to define these regions is the ratio between the production rate of hydrogen peroxide (H$_2$O$_2$) and nitric acid (HNO$_3$).





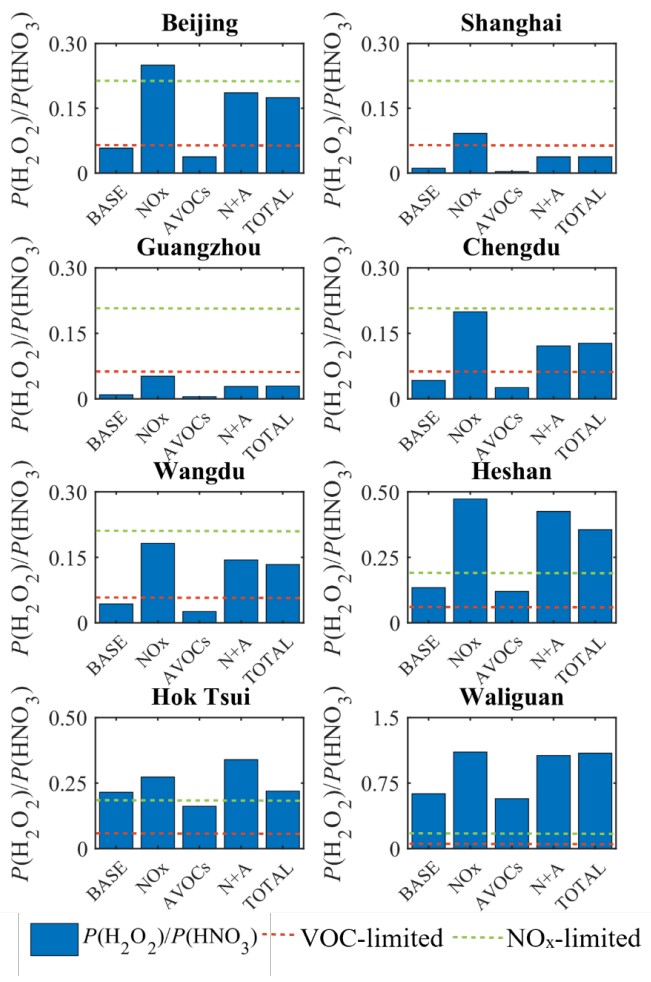

Figure 2. The daytime (06:00 to 19:00 Local Standard Time (LST)) value of ratio between the production rate of hydrogen peroxide ($H_2O_2$) and nitric acid ($HNO_3$) [$P(H_2O_2)/P(HNO_3)$] in different regions of China for July 2018. The value of [$P(H_2O_2)/P(HNO_3)$] below the dotted line in red (0.06), above the dotted line in green (0.2), and in between represents the control by VOCs, $NO_x$, and in transition.



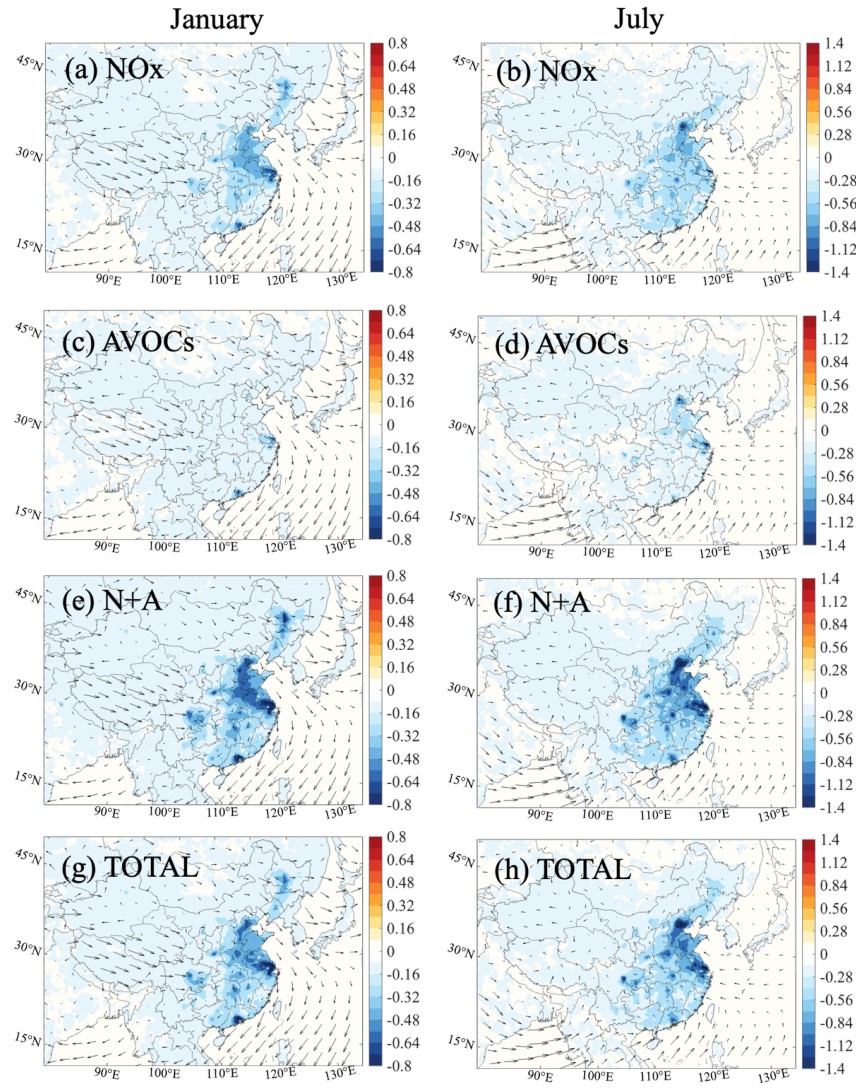

Figure 3. Changes in the averaged daytime surface production rate of $RO_x$ ($RO_2$+$HO_2$+OH) [Unit: ppbv $h^{-1}$] response to a 50% reduction in $NO_x$ emissions (a, b; NOx case), in anthropogenic VOCs (AVOCs) emissions (c, d; AVOCs case), in $NO_x$ and AVOCs emissions (e, f; N+A case), and in all anthropogenic emissions (g, h, TOTAL case) relative to BASE case. Results are shown for January (a, c, e, g) and July (b, d, f, h) of 2018. Arrows represent the wind speed and wind direction.



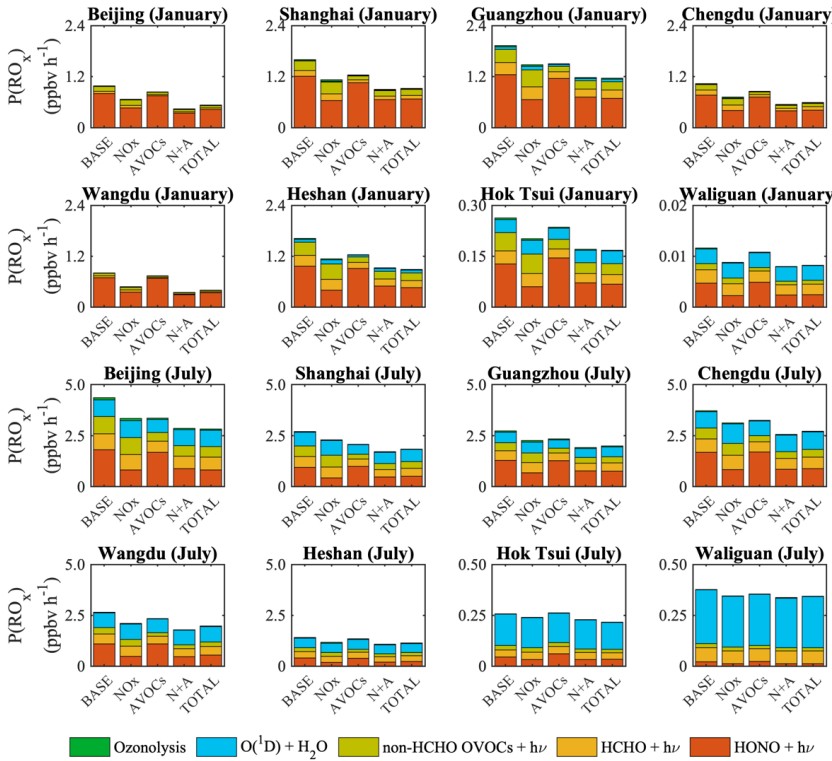

Figure 4. Averaged daytime value of production rate of $RO_x$ ($P(RO_x)$) [Unit: ppbv h$^{-1}$] in five different simulated cases (BASE, NOx, AVOCs, N+A, TOTAL cases) and eight different sites (urban, rural, and remote sites) in January and July of 2018. The two upper rows refer to January and the two lower rows to July.



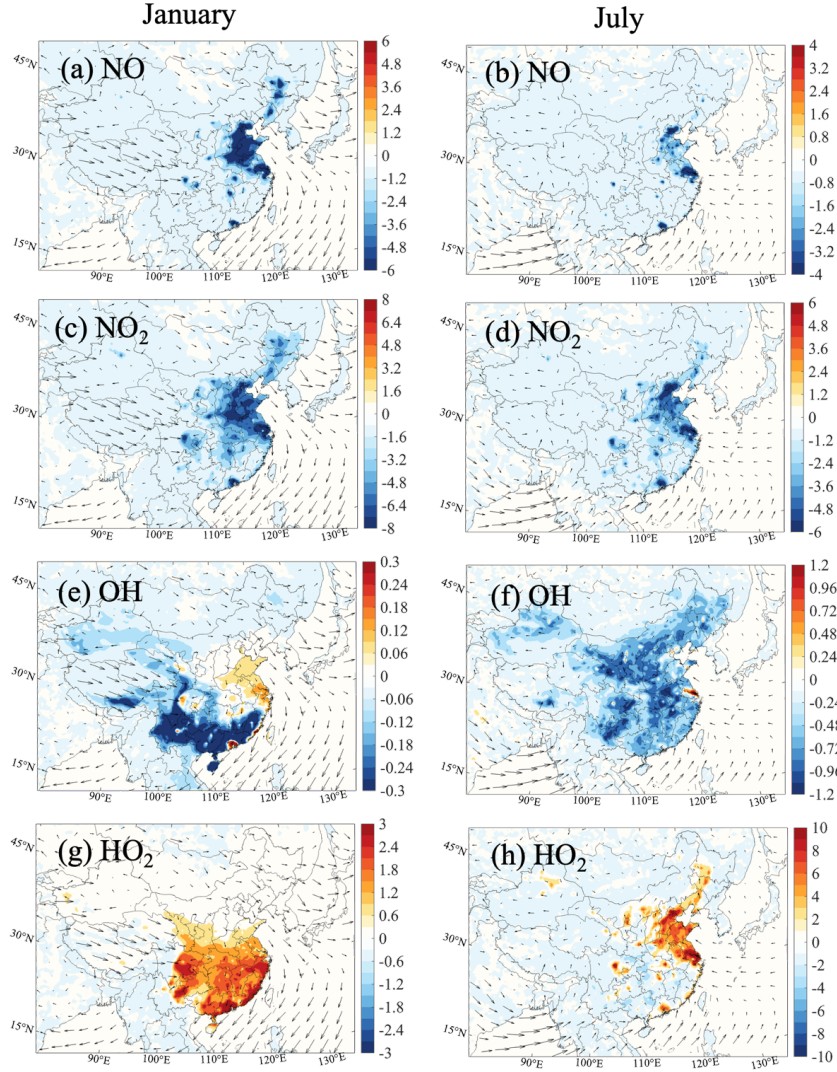


Figure 5. Changes in the surface mixing ratio of NO (a, b) [Unit: ppbv], $NO_2$ (c, d) [Unit: ppbv], OH radical (e, f) [Unit: 0.1 pptv] and $HO_2$ radical (g, h) [Unit: pptv] response to the NOx case relative to BASE case. The results are shown for January (a, c, e, g) and July (b, d, f, h) of 2018. Arrows represent
the wind speed and wind direction.



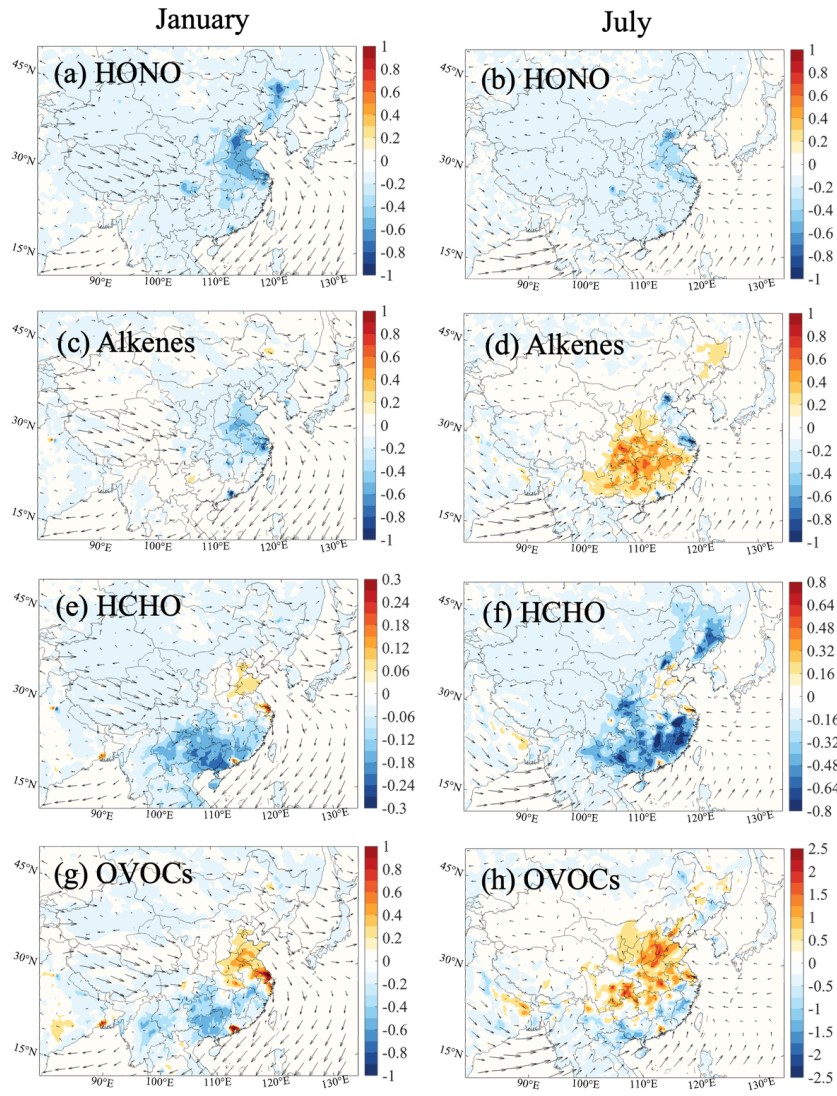

Figure 6. Changes in the surface mixing ratio of HONO (a, b) [Unit: ppbv], alkenes (c, d) [Unit: ppbv], formaldehyde (HCHO; e, f) [Unit: ppbv] and total oxidized VOCs (OVOCs; g, h) [Unit: ppbv] response to the NOx case relative to the BASE case. The results are shown for January (a, c, e, g) and July (b, d, f, h) of 2018. Arrows represent the wind speed and wind direction.



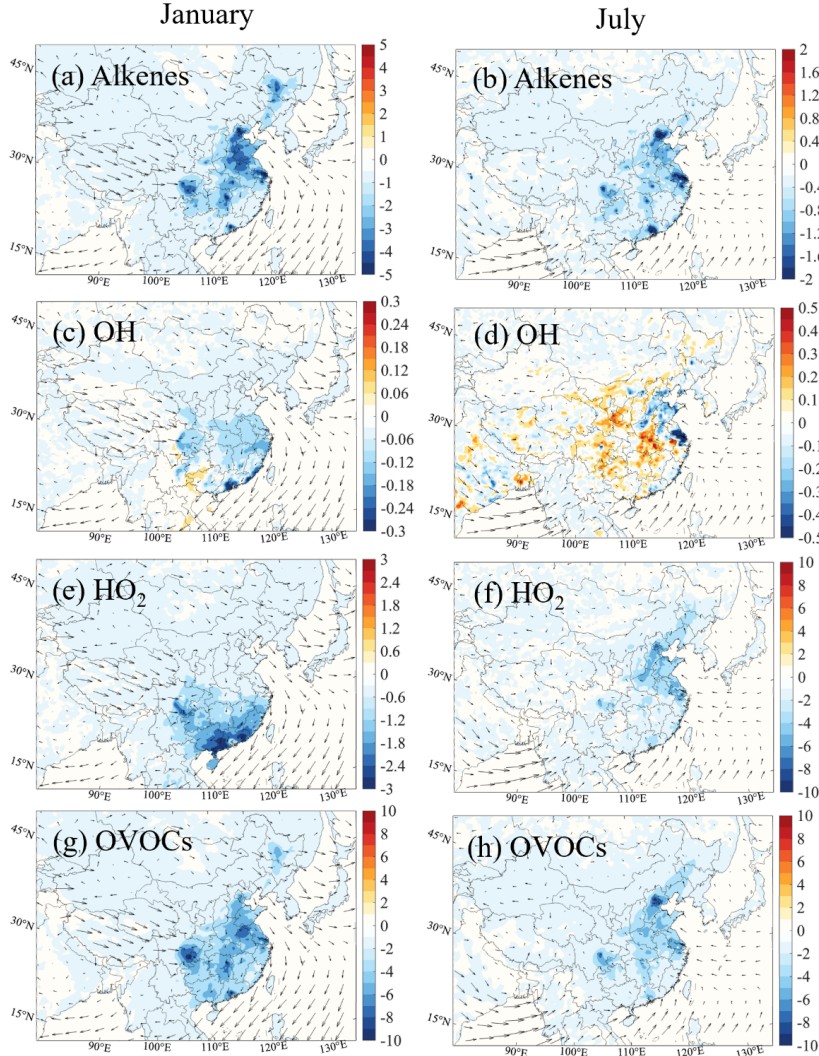


Figure 7. Changes in the surface mixing ratio of alkenes (a, b) [Unit: ppbv], OH radical (c, d) [Unit: 0.1 pptv], HO$_2$ radical (e, f) [Unit: pptv], and OVOCs (g, h) [Unit: pptv] response to the ratio of 0.5 in AVOCs emissions (AVOCs case) relative to BASE case. Results are shown for January (a, c, e, g) and July (b, d, f, h) of 2018. Arrows represent the wind speed and wind direction.




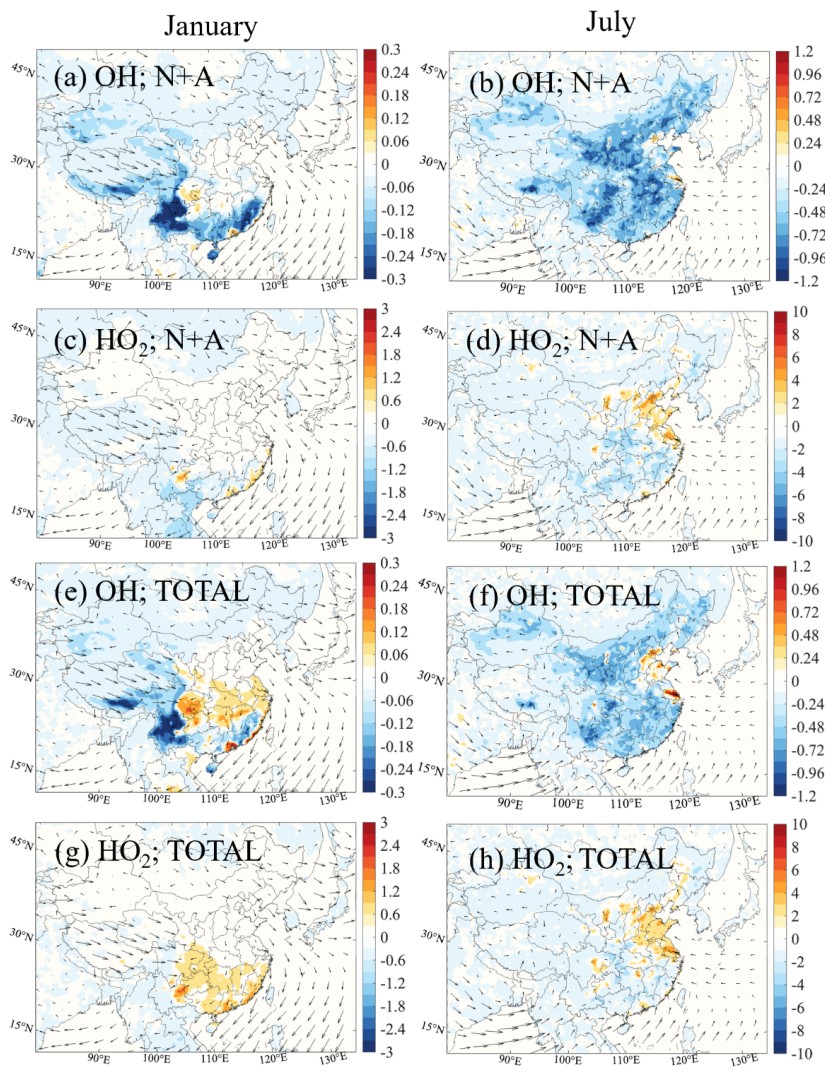


Figure 8. Changes in the surface mixing ratio of OH radical (a, b, e, f) [Unit: 0.1 pptv] and HO$_2$ radical (c, d, g, h) [Unit: pptv] response to the ratio of 0.5 in NO$_x$ and AVOCs emissions (N+A case) and in all anthropogenic emissions (TOTAL case) relative to BASE case. Results are shown for January (a, c, e, g) and July (b, d, f, h) of 2018. Arrows represent the wind speed and wind direction.





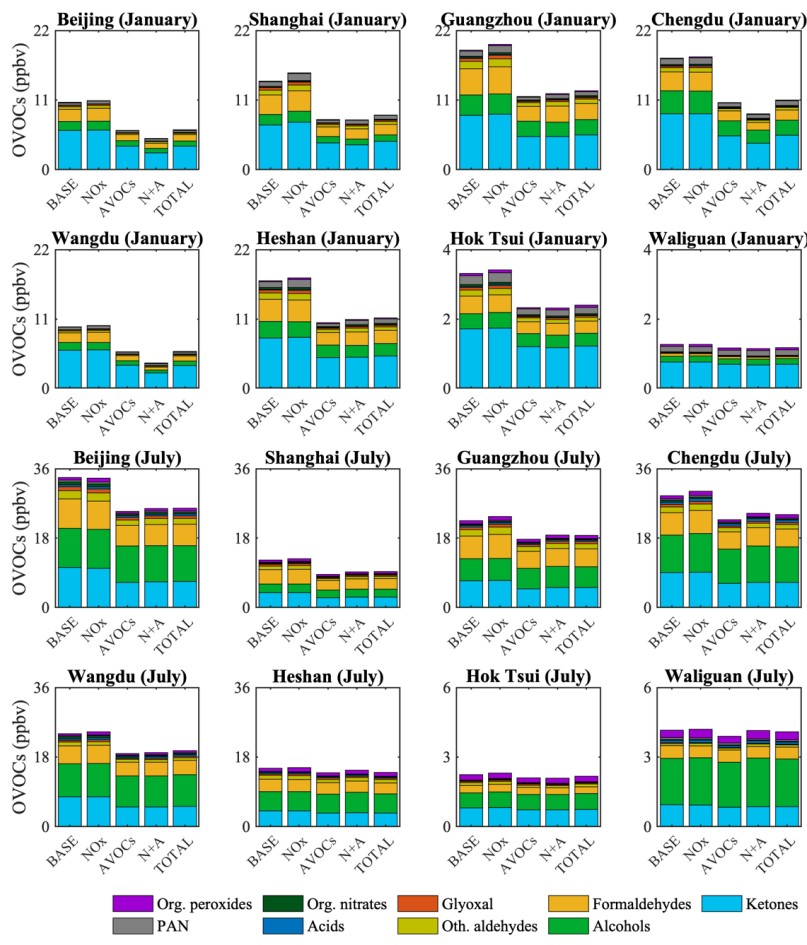

Figure 9. Averaged mixing ratio of oxidized VOCs (OVOCs; Unit: ppbv) with the contribution
from nine types of species in five simulated cases (BASE, NOx, AVOCs, N+A, TOTAL cases)
and at eight sites (urban, rural, and remote sites) in January and July of 2018. The two upper
rows refer to January and the two lower rows to July.



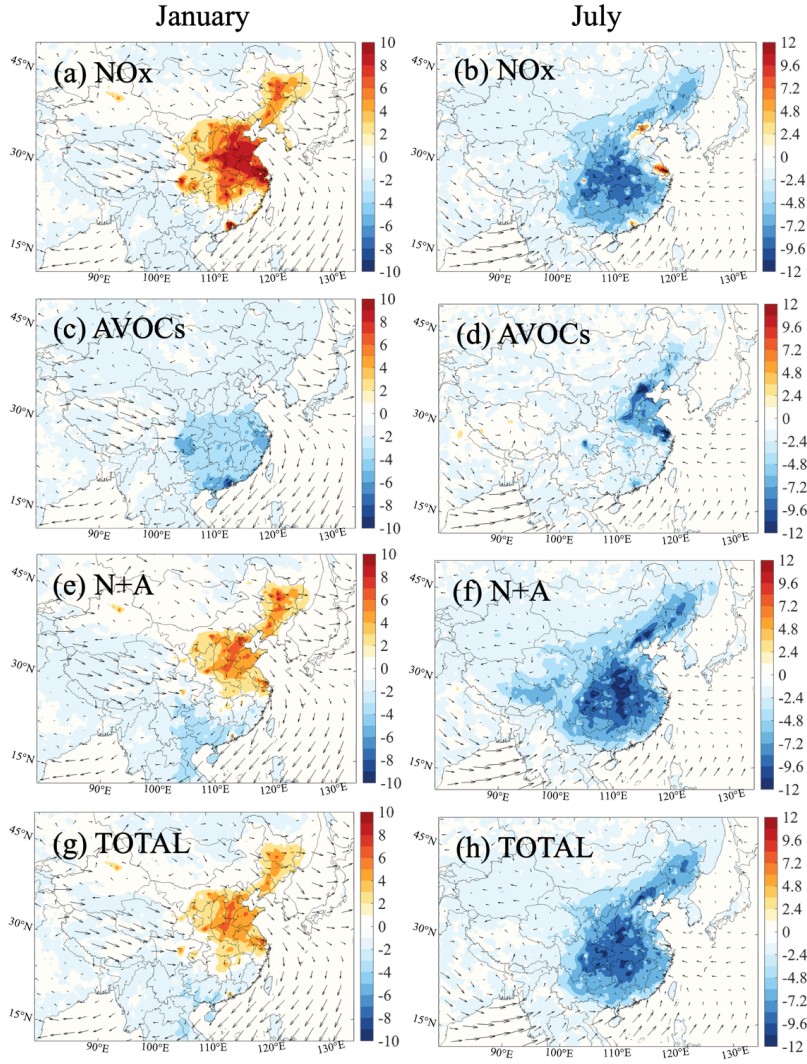


Figure 10. Changes in the averaged daytime surface mixing ratio of ozone [Unit: ppbv] response to the NOx case (a, b), AVOCs case (c, d), N+A case (e, f), and TOTAL case (g, h) relative to BASE case. The results are shown for January (a, c, e, g) and July (b, d, f, h) of 2018. Arrows represent the wind speed and wind direction.




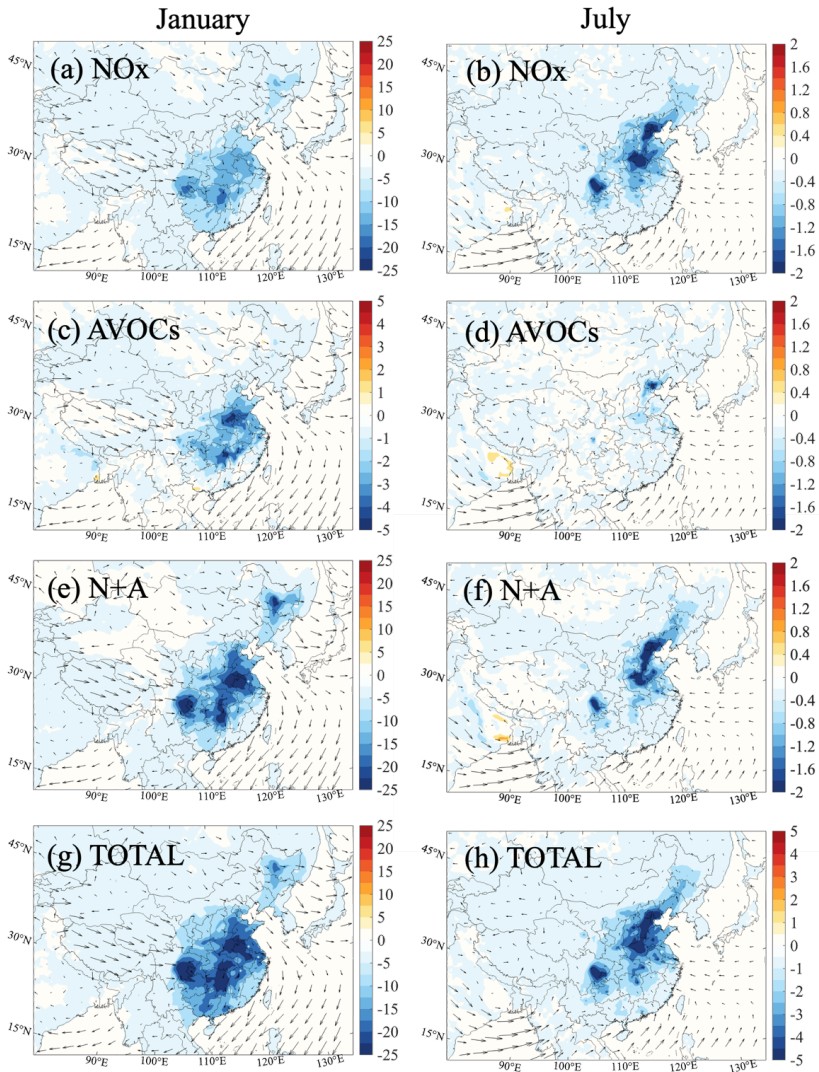

Figure 11. Changes in the surface concentration of Secondary Inorganic Aerosol [SIA, Unit: μg m⁻³] response to NOx case (a, b), AVOCs case (c, d), N+A case (e, f), and TOTAL case (g, h) relative to BASE case. Results are shown for January (a, c, e, g) and July (b, d, f, h) of 2018. SIA represents the sum of nitrate, sulfate, and ammonia. Notice the inconsistency in the scale of Figure 11c and h. Arrows represent the wind speed and wind direction.

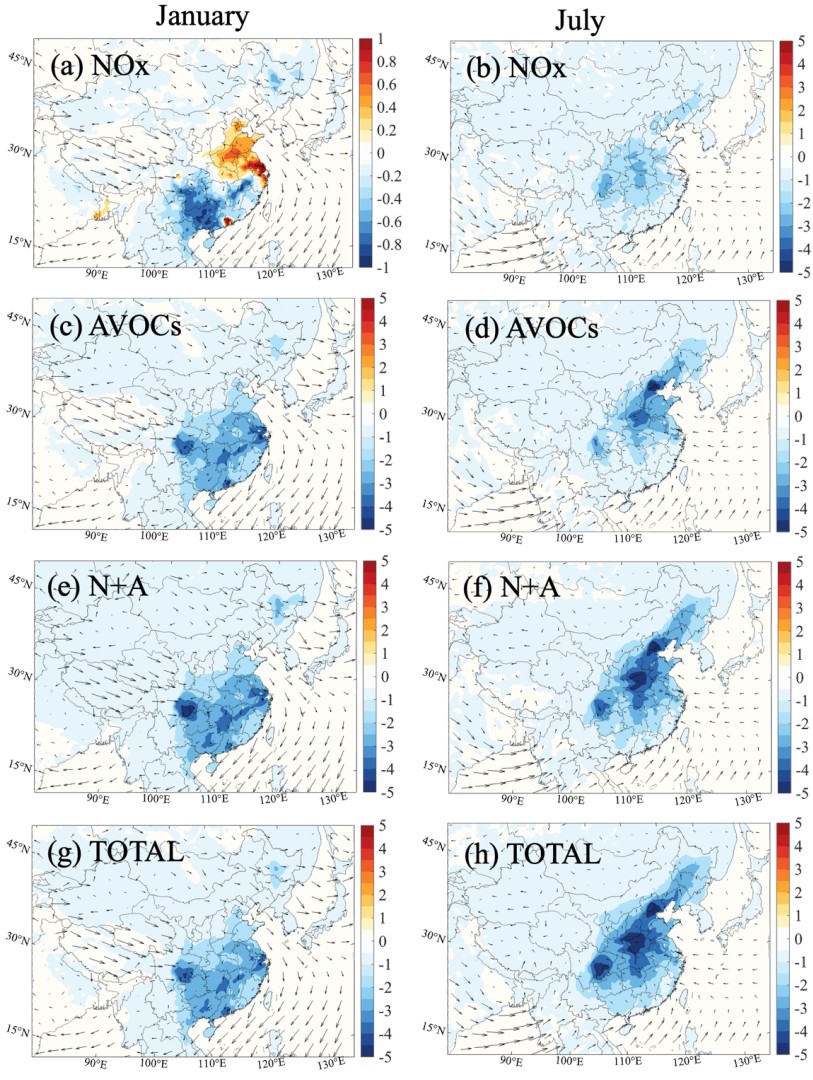


Figure 12. Changes in the surface concentration of Secondary Organic Aerosol [SOA, Unit: $\mu g\ m^{-3}$] response to NOx case (a, b), AVOCs case (c, d), N+A case (e, f), and TOTAL case (g, h) relative to BASE case. Results are shown for January (a, c, e, g) and July (b, d, f, h) of 2018.
Notice the inconsistency in the scale of Figure 12a. Arrows represent the wind speed and wind direction.

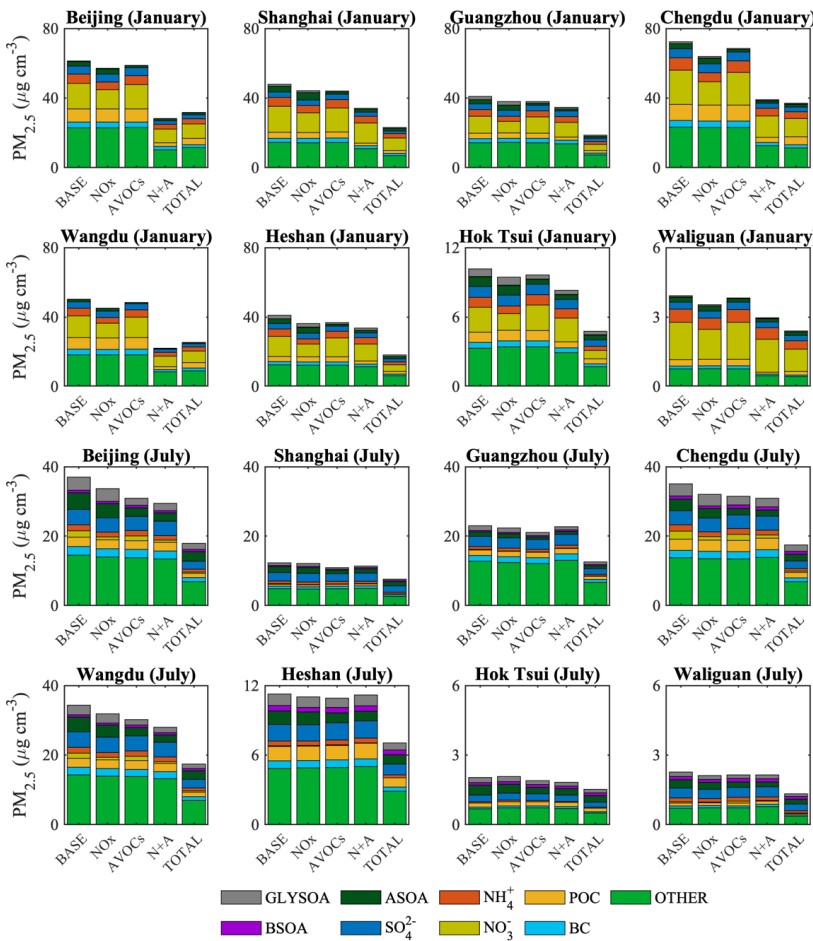

Figure 13. Averaged concentration of PM$_{2.5}$ [Unit: μg m$^{-3}$] with the contribution from different chemical compositions in five simulated cases and in eight sites for July 2018. The fine particle is composed of particulate nitrate (NO$_3^-$), particulate sulfate (SO$_4^{2-}$), particulate ammonia (NH$_4^+$), primary organic carbon (POC), black carbon (BC), anthropogenic secondary organic aerosol (ASOA), biogenic secondary organic aerosol (BSOA), secondary organic aerosol from glyoxal (GLYSOA), and other aerosol compositions (OTHER, including sea salt, carbonate, calcium, minerals, and other inorganic mass). The two upper rows refer to January and the two lower rows to July.



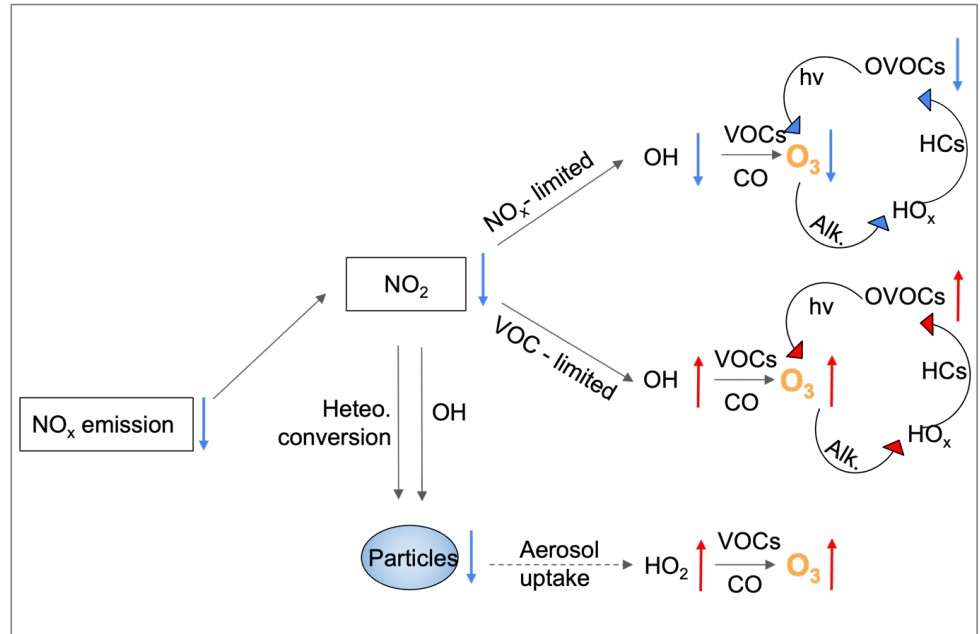

Figure 14. Schematics for the impact of $NO_x$ emission reduction through aerosol effect and photochemical processes on ozone concentration. Arrows represent the changes in the concentration of chemicals associated with the reduction of $NO_x$ emission (decrease trend shown in blue increase trend shown in red). HCs and Alk. are the abbreviations of hydrocarbons and alkenes.



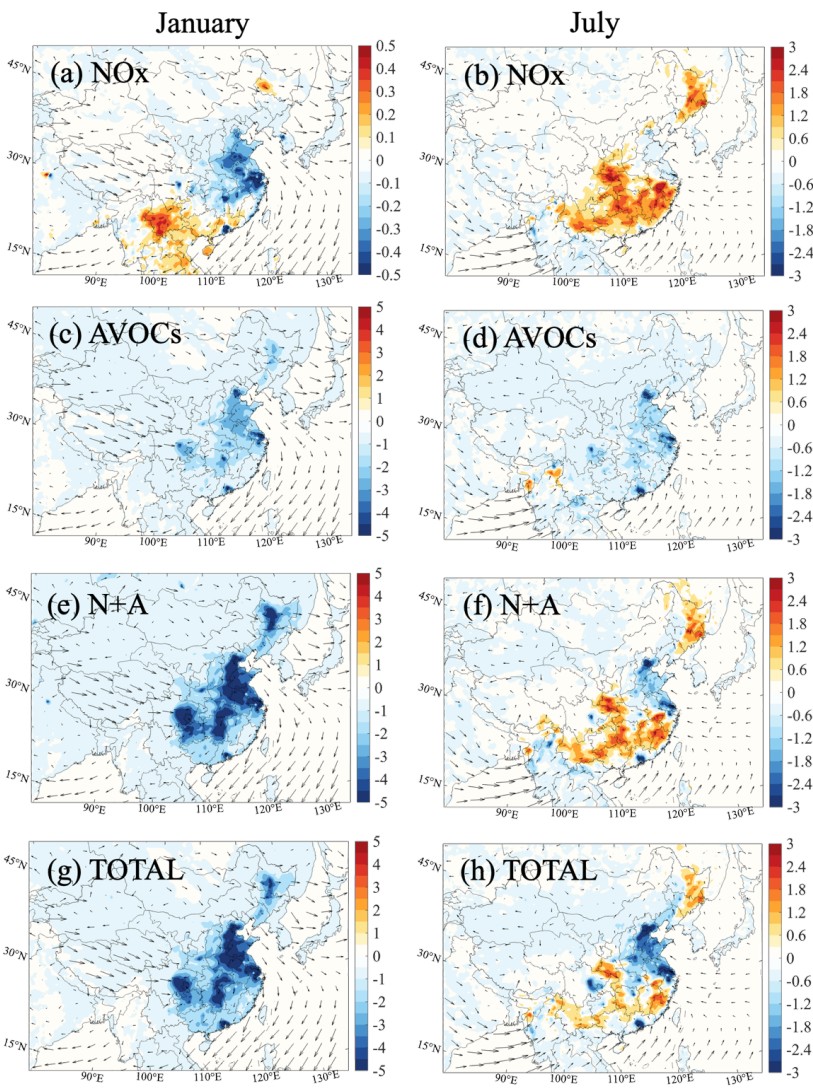


Figure 15. Changes in the average daytime OH reactivity from VOCs and CO [Unit: s⁻¹] response to the NOx case (a, b), AVOCs case (c, d), N+A case (e, f), and TOTAL case (g, h) relative to BASE case. Results are shown for January (a, c, e, g) and July (b, d, f, h) of 2018. Arrows represent the wind speed and wind direction.





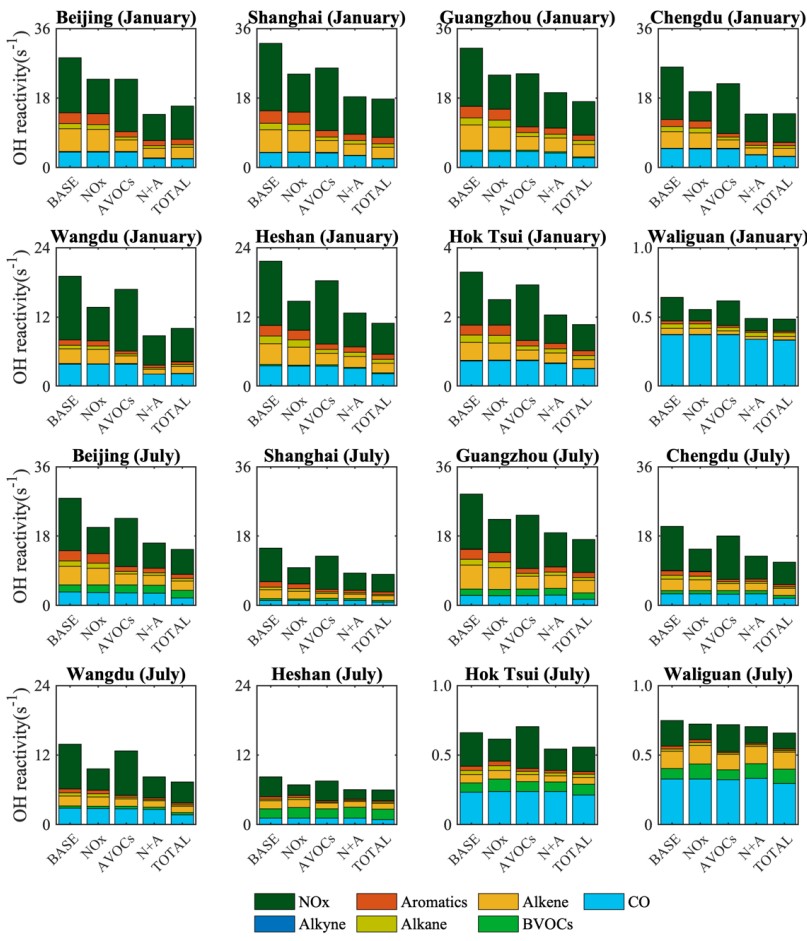

Figure 16. Averaged value of daytime OH reactivity [Unit: $s^{-1}$] with the contribution from seven different species in five different simulated cases (BASE, NOx, AVOCs, N+A, TOTAL cases) and in eight different sites in July. The two upper rows refer to January and the two lower rows to July.




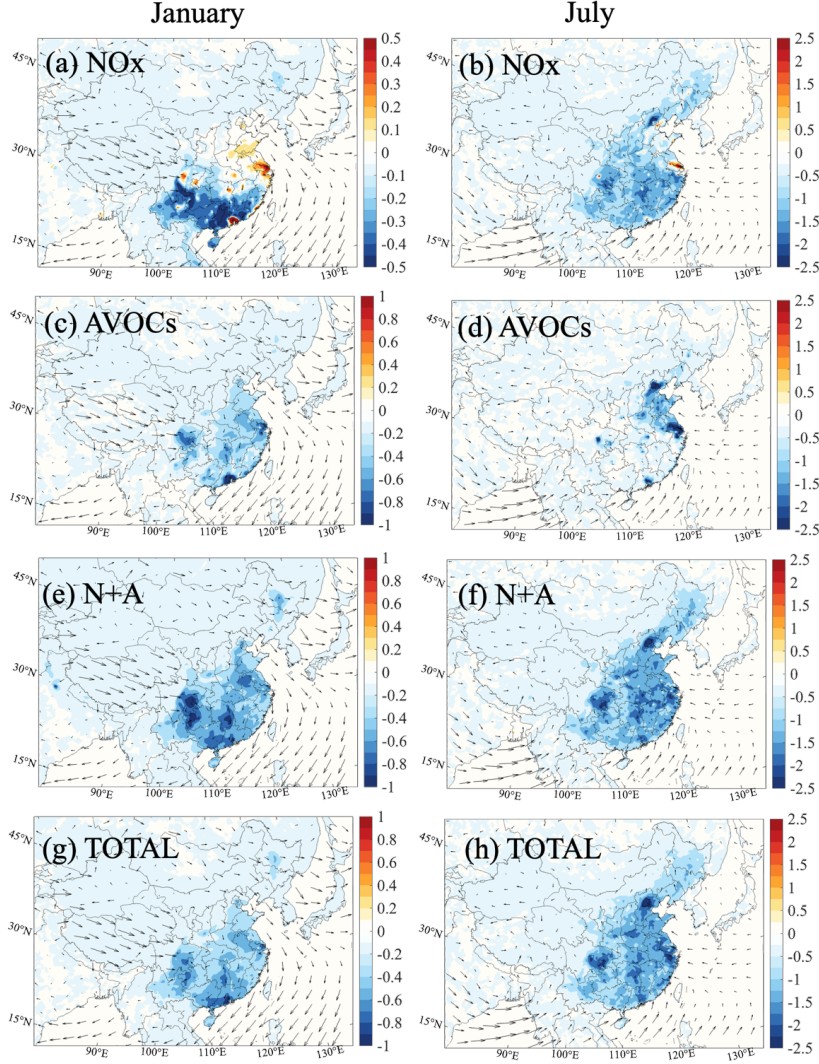

Figure 17. Changes in the daytime averaged atmospheric oxidizing capacity ($AOC$) [Unit: $10^7$ molec. cm$^{-3}$ s$^{-1}$] response to the NOx case (a, b), AVOCs case (c, d), N+A case (e, f), and TOTAL case (g, h) relative to BASE case. Results are shown for January (a, c, e, g) and July (b, d, f, h) of 2018. Arrows represent the wind speed and wind direction.



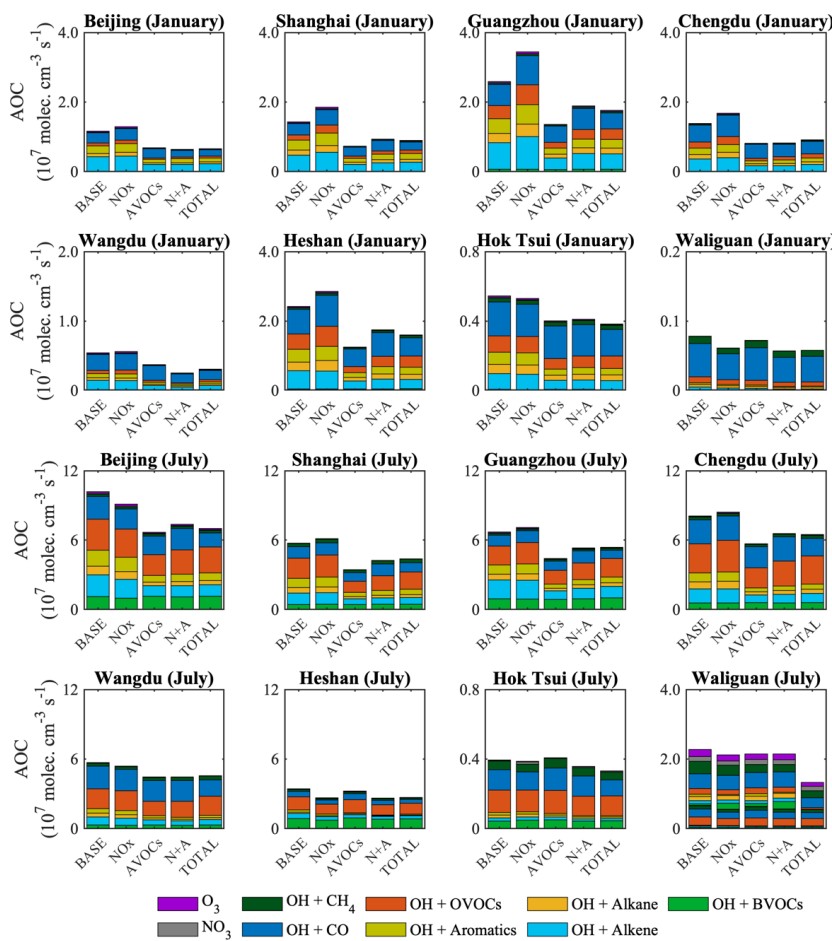

Figure 18.  Averaged value of *AOC* [Unit: $10^7$ molec. cm$^{-3}$ s$^{-1}$] during daytime in nine different
species in five different simulated cases (BASE, NOx, AVOCs, N+A, TOTAL cases) and in
eight different sites in July. The two upper rows refer to January and the two lower rows to
July.



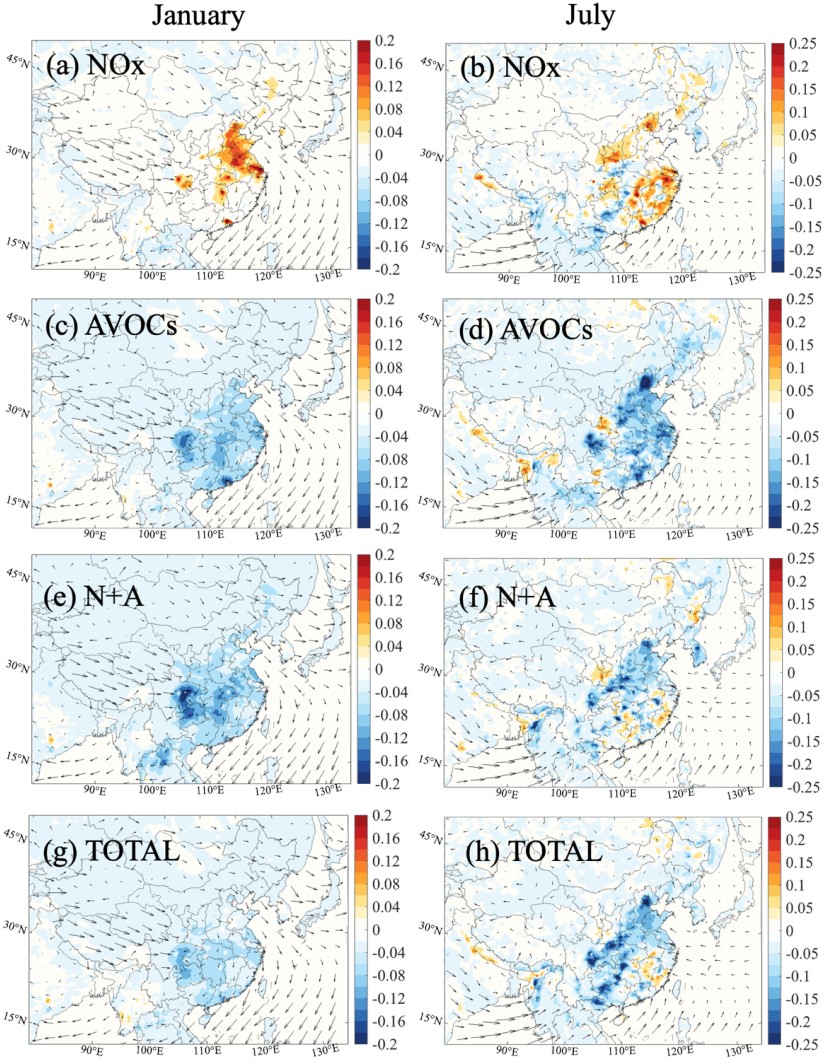

Figure 19. Spatial distribution of the averaged nighttime (20:00 to 05:00 LST) atmospheric oxidizing capacity (*AOC*) due to the reactions between ozone and alkenes [Unit: $10^6$ molec. cm$^{-3}$ s$^{-1}$] response to the ratio of 0.5 in NO$_x$ emissions (a, b; NOx case), in Anthropogenic VOCs (AVOCs) emissions (c, d; AVOCs case), in NO$_x$ and AVOCs emissions (e, f; N+A case), and all anthropogenic emissions (g, h, TOTAL case) relative to BASE case. Results are shown for January (a, c, e, g) and July (b, d, f, h) of 2018. Arrows represent the wind speed and wind direction.





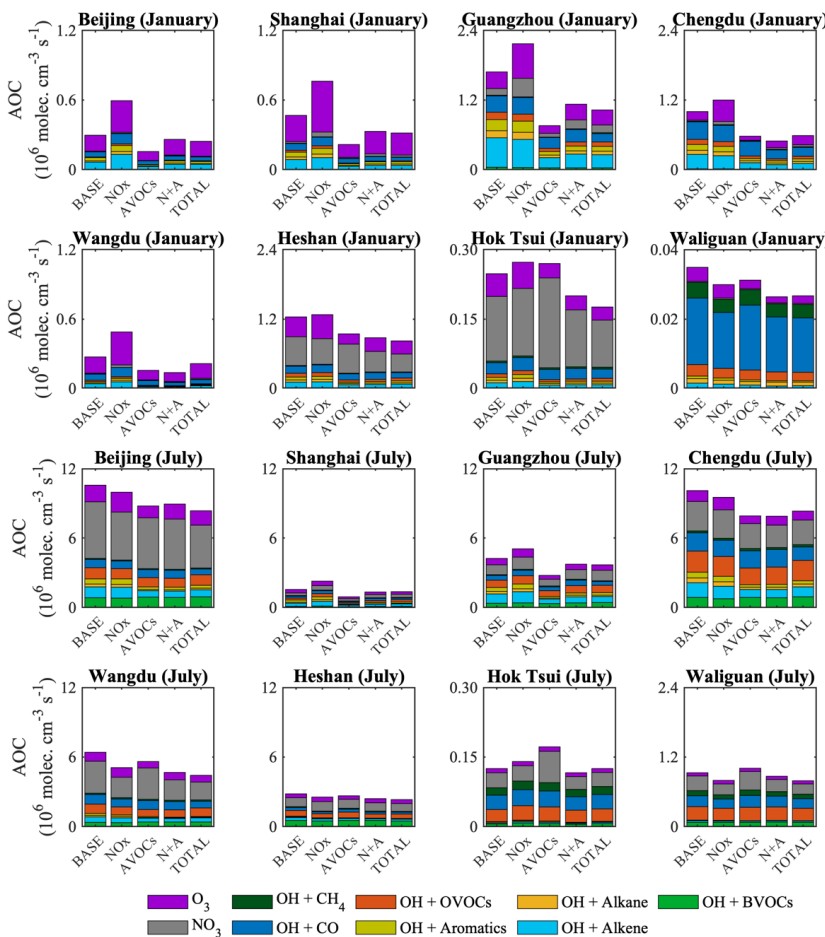


Figure 20.  Averaged value of *AOC* [Unit: $10^6$ molec. cm$^{-3}$ s$^{-1}$] during nighttime in nine different species in five different simulated cases (BASE, NOx, AVOCs, N+A, TOTAL cases) and in eight different sites in July. The two upper rows refer to January and the two lower rows to July.
