# Peer review of "The Atmospheric Oxidizing Capacity in China: Part 2. Sensitivity to emissions of primary pollutants"

_EGUsphere, 2024_

## Referee Comment (RC1)

**General Comments**

This paper contains very lengthy descriptions of model responses to changes in emissions of NOx, VOCs, and both in China. The goal of the paper is to help policy-makers mitigate ozone increases in urban areas. However this message gets lost in the lengthy descriptions throughout the paper. The authors should consider significantly shortening their descriptions with a focus on how their findings can clarify the impact of policy measures that either reduce NOx, VOCs, or both. Do we need a description of changes in $pRO_x$, $pO_x$, OH reactivity, $NO_x$ reactivity, and AVOC? Could a subset of the plots provided with a focus on high ozone and $PM_{2.5}$ areas suffice? What is the message the authors want to give regarding $PM_{2.5}$?

The authors should also better explain their finding that the combined reduction of NOx and AVOC emissions has a larger effect on both ozone and PM2.5 than the sum of the reductions of NOx and AVOC separately. Currently, it reads for example that the best 'value' to be gained in reducing P(Ox) in summer is reducing NOx. This is also said on line 390. Overall, the authors have too much detail on specific changes in their model, and insufficient description of the broader new understanding gained or policy-insights developed.

The authors have a modeling setup that could provide insight into the benefits of different types of emissions reductions and help us gain insight into the impacts of the reductions in aerosol on ozone concentrations. The aerosol impacts on ozone could be the most interesting part of this paper but the manuscript as written is far too lengthy and lacks clear and concise messaging. The authors describe many model metrics (pROx, pOx, AVOC) but it is not clear what different insights are gained from each one, or if a singular metric would suffice to describe the relevant model impacts. If the authors are able to revise the paper to increase the value of their scientific analysis and refine their messaging, then it would be appropriate for publication.

**Specific Comments**

Intro: I would expect that in VOC-limited areas, decreasing $NO_x$ would result in higher OH from reduced loss of OH to OH + $NO_2$ → HNO3. Thus, $HO_2$ would be higher from increased VOC oxidation. Is the aerosol uptake effect on $NO_2$ from reduced nitrate aerosol really larger?

Line 110 – "nitration" should be "titration". Also, please clarify the meaning of this statement "and the competition between $NO_2$ and VOC for OH radicals".

Line 214 – "Validated" implies the model was correct in the companion study while there were a variety of model shortcomings described such as overestimated summertime $NO_2$ and $PM_{2.5}$. It would be better to describe how any model biases impact the conclusions rather than call the model "validated".

Line 237 – Has anyone done a weekend/weekday analysis of ozone to see whether ozone goes up or down when NOx is reduced on the weekends, assuming that is the case in China? A quick search found studies like this: https://www.sciencedirect.com/science/article/pii/S1474706518302110, or https://www.nature.com/articles/s41598-020-64111-3. If so, please cite those studies here as support for your spatial distribution of regimes.

Line 312 – Can you explain why that is?

Section 3.2: This goes into great detail on how the budgets of radicals change, and I find it difficult to see what the overall conclusion is that is either policy-relevant or novel. Instead, it just reads like a helpful description of the model behavior which may be useful for other modelers but is not necessary in the main text. In that case, the paper could be shortened by a quick summary of the major effects (less NOx = less loss to HNO3, less VOCs = less pROx from OVOCs, less CO = more pROx due to higher OH etc) and moving the majority of the discussion to the supplement. If not, the authors need to better state the importance of their description.

Line 390 – Can the authors be more specific about the meaning of "further enhanced"? Why should policy makers bother if most of the impact is from $NO_x$?

Line 392 – The authors state that reductions in 'specific AVOCs' are needed but so far they have only discussed AVOCs as a whole.

Line 424 – The increase in OH in Hong Kong appears very strong as well.

Line 473 – Does the model really have NH3 + OH as a significant sink of OH? If the authors are referring to its impact on SIA and thus $HO_2$ uptake, this is not clear.

Paragraph starting on line 567 – This discussion is again very lengthy. The figure appears to show that the most important message is that $NO_x$ reductions in July alone result in ozone decreases in several major cities (Beijing, Shanghai, Hong Kong?) while adding in AVOC reductions causes the cities to also see an ozone decrease.

Line 599 – How does reduced AVOCs impact nitrate, sulfate, and ammonium? It is not clear from this sentence.

Line 628 – How much does photolysis increase in your model with reduced aerosol?

Figure 13 – What is in 'Other' that is impacted in your 'TOTAL' case? This category is a surprisingly large fraction of model $PM_{2.5}$ and thus deserves more discussion. Overall, Fig. 13 contains a lot of information but is barely discussed.

Line 645 – Why have a schematic for reduction in $NO_x$ emissions, but not AVOC emissions, and the combination of the two?

Section 3.4.1 – Again, I am not sure what the main message is from this lengthy section.

Line 755 – What is the result of the increased ozonolysis? Do we get more OVOCs that impact daytime air quality? This is said later but is not clear here.

Line 771 – Are these primary OVOCs like methanol or ethanol? Or secondary species like HCHO and acetaldehyde? If secondary, then what are their main precursors? Which 'unsaturated OVOCs' should be targeted?

Summary – Again, a greater focus on policy-relevant insights would be helpful as there are opposite effects on average compared to in the major cities.

Line 795 – What about reduced loss of OH to OH + $NO_2$ which increases the ability to oxidize VOCs?

Line 830 – The reason for the greater joint impact needs to be explained.

Line 853 – Refrain from discussing 'slight' changes to focus on the major findings.

Line 869 – Here the authors state that their goal is to help develop a strategy for metropolitan areas. If this is the goal of the paper, the authors should consider a greater focus on the impacts on cities (bar chart figures such as Fig. 4).

Line 869 – The authors already specifically call out categories of VOCs (alkenes, aromatics etc). Could the authors better describe what they mean by 'more detailed investigations' here?

Code and data availability: This does not include the modifications made to WRF-Chem described in Dai et al., 2023 and used here.

---

## Author Comment (AC1)

**Response to Reviewers' Comments**

**Dear Editor and Reviewers,**

Thank you very much for your efforts in handling and evaluating our submission.
The review comments are very helpful for improving the original manuscript. We have carefully considered them and tried to address all of these comments in the revised version of the manuscript. Below are the detailed point-by-point responses to the review comments. For clarity, the reviewer's comments are listed below in *black italics*, while our responses and changes in the manuscript are highlighted in blue and red, respectively.
We look forward to receiving a further evaluation of our work.

Best regards,
Guy Brasseur and co-authors

**General Comments**

*This paper contains very lengthy descriptions of model responses to changes in emissions of NOx, VOCs, and both in China. The goal of the paper is to help policymakers mitigate ozone increases in urban areas. However, this message gets lost in the lengthy descriptions throughout the paper. The authors should consider significantly shortening their descriptions with a focus on how their findings can clarify the impact of policy measures that either reduce NOx, VOCs, or both. Do we need a description of changes in pROx, pOx, OH reactivity, NOx reactivity, and AVOC? Could a subset of the plots provided with a focus on high ozone and PM2.5 areas suffice? What is the message the authors want to give regarding PM2.5?*

*The authors should also better explain their finding that the combined reduction of NOx and AVOC emissions has a larger effect on both ozone and PM2.5 than the sum of the reductions of NOx and AVOC separately. Currently, it reads for example that the best 'value' to be gained in reducing P(Ox) in summer is reducing NOx. This is also said on line 390. Overall, the authors have too much detail on specific changes in their model, and insufficient description of the broader new understanding gained or policy-insights developed.*

*The authors have a modeling setup that could provide insight into the benefits of different types of emissions reductions and help us gain insight into the impacts of the reductions in aerosol on ozone concentrations. The aerosol impacts on ozone could be the most interesting part of this paper but the manuscript as written is far too lengthy and lacks clear and concise messaging. The authors describe many model metrics (pROx, pOx, AVOC) but it is not clear what different insights are gained from each one, or if a singular metric would suffice to describe the relevant model impacts. If the authors are able to revise the paper to increase the value of their scientific analysis and refine their messaging, then it would be appropriate for publication.*

Author's reply: We would like to thank the reviewer for so carefully reading our manuscript and for making all the comments. Based on these comments, we significantly condensed our paper, which was judged to be lengthy. We have condensed the description of our model results and have focused on providing insights about how to mitigate ozone increase in urban China. Regarding the metrics used in this paper to characterize the photochemical environment, we only kept the changes in the Atmospheric Oxidative Capacity (AOC) in the main text and moved a reference to the other metrics to supplement. Regarding PM$_{2.5}$, we largely shortened the description on the changes in PM$_{2.5}$ due to emissions reduction and elaborated on how the changes the in the aerosol load affect the ozone formation. The structure of the new version of the paper is expressed in the paper as follows (Line 158-166 in Text).

"This paper is structured as follows. Section 2 introduces the setups of the model system and describes the simulations performed for specified reductions in the emissions of

primary pollutants. In Section 3, we first analyze the response in the near-surface concentration of ozone precursors and intermediate to primary emission reduction. We also discuss the changes in the ozone formation regime. Further, we derive the associated changes in ozone, and aerosols to emission reductions. Finally, we describe the sensitivity of the atmospheric oxidative capacity (AOC) to the reduction in emissions. A summary and implication for policy making of our study is provided in Sec. 4."

**Specific Comments**

*Intro: I would expect that in VOC-limited areas, decreasing NOx would result in higher OH from reduced loss of OH to OH + NO2 to HNO3. Thus, HO2 would be higher from increased VOC oxidation. Is the aerosol uptake effect on NO2 from reduced nitrate aerosol really larger?*

Author's reply: Based on our results and regarding the increased concentration of $HO_2$, it is difficult to compare the contribution of the VOC oxidation by enhanced OH with the contribution of a reduced aerosol uptake. To clarify the underlying reasons for the summertime ozone increase, we changed the sentence in the introduction as follows (Line 55-58 in Text).

"This $O_3$ increase is associated with a reduced $NO_x$-titration effect and with higher levels of hydroxyl (OH) and hydroperoxyl ($HO_2$) radicals due to a reduced loss by reactions with nitrogen dioxide ($NO_2$) and by a decreased aerosol uptake".

*Line 110 – "nitration" should be "titration". Also, please clarify the meaning of this statement "and the competition between $NO_2$ and VOC for OH radicals".*

Author's reply: We corrected the mistake and deleted the unclear statement. The sentence is changed to (Line 111-113 in Text):

"In VOC-sensitive regimes, the reduction in the NOx abundance tends to enhance the ozone formation due to the weakening of NO titration and the reduced loss of OH radical reacted with $NO_2$".

*Line 214 – "Validated" implies the model was correct in the companion study while there were a variety of model shortcomings described such as overestimated summertime NO2 and PM2.5. It would be better to describe how any model biases impact the conclusions rather than call the model "validated".*

Author's reply: We added one sentence in the text, regarding the discussion of the aerosol effect on the ozone formation (Line 504-508 in Text):

"An overestimation in the concentration of $NO_2$ and $PM_{2.5}$ has been simulated for the

baseline conditions, which can possibly lead to a higher reduction in aerosol concentration, especially in the concentration of $NO_3^-$. This overestimation potentially affects the aerosol-related changes in ozone formation."

*Line 237 – Has anyone done a weekend/weekday analysis of ozone to see whether ozone goes up or down when NOx is reduced on the weekends, assuming that is the case in China? A quick search found studies like this: https://www.sciencedirect.com/science/article/pii/S1474706518302110, or https://www.nature.com/articles/s41598020-64111-3. If so, please cite those studies here as support for your spatial distribution of regimes.*

Author's reply: These two references as well as the reference to (Tonnesen and Dennis, 2000) are added to our manuscript to support our description on the ozone sensitivity regimes.

*Line 312 – Can you explain why that is?*

Author's reply: The higher value of $P(RO_x)$ in the TOTAL case (with the emissions reduction in $NO_x$, AVOCs, $NH_3$, $SO_2$, and CO) relative to the level in the N+A case (with the emission reduction in $NO_x$ and AVOCs) is due to the higher concentration of OVOCs (Figure 1a) and of ozone (Figure 1b), whose photolysis produces more photochemical radicals.

[Figure]

Figure 1. Changes in the concentration of OVOCs and ozone at the surface due to the emissions reduction in TOTAL case relative to N+A cases for January 2018.

*Section 3.2: This goes into great detail on how the budgets of radicals change, and I find it difficult to see what the overall conclusion is that is either policy-relevant or novel. Instead, it just reads like a helpful description of the model behavior which may be useful for other modelers but is not necessary in the main text. In that case, the paper could be shortened by a quick summary of the major effects (less NOx = less loss to HNO3, less VOCs = less pROx from OVOCs, less CO = more pROx due to higher OH*

*etc) and moving the majority of the discussion to the supplement. If not, the authors need to better state the importance of their description.*

Author's reply: Following your suggestions, we deleted the details dealing with the description of the changes in the radical's budget and condensed it with the description of the changes in the surface mixing ratios OH and HO$_2$ radicals in response to emissions reduction (in Section 3.1.1).

*Line 390 – Can the authors be more specific about the meaning of "further enhanced"? Why should policy makers bother if most of the impact is from NOx?*

Author's reply: Based on our results, the impact of NO$_x$ emissions reduction on the production of radicals is larger than the impact of AVOCs emissions reduction. The decrease in radical due to reduced NO$_x$ emissions can be partially counteracted by the reduced AVOCs emissions in the urban areas. As shown in Figure 2a, the reduction in NO$_x$ leads to a decrease in the OH radical of the non-urban areas (NOx-limited) in southern China, while an increase in the OH radical in urban areas (VOC-limited), results from the reduced loss by the reaction with reduced NO$_2$. The reduction in AVOCs leads to decreases in the OH radicals mixing ratio (Figure 2b), due to the reduced VOC oxidation. As the reduction in NO$_x$ emission alone is not sufficient for reducing the concentration of the OH radical in all geographical areas in China, a concomitant reduction in AVOCs emissions is needed. More details can be found in Section 3.1.1 of our paper.

[Figure]

Figure 2. Changes in the averaged daytime surface mixing ratio of the OH radical (a-c, Unit: 0.1 pptv) in response to a 50% reduction in NO$_x$ emissions (a; NOx case), in anthropogenic VOCs (AVOCs) emissions (b; AVOCs case) and in NO$_x$ and AVOCs emissions (c; N+A case) relative to the BASE case for January of 2018.

*Line 392 – The authors state that reductions in 'specific AVOCs' are needed but so far they have only discussed AVOCs as a whole.*

Author's reply: In our simulations for the present study, we only considered the reduction in AVOCs emissions as a whole without partitioning between species. However, we discussed the impact of AVOCs emissions reduction on specific VOCs (hydrocarbons and OVOCs), listed in Section 3.1.2; and the changes in the contribution

of the reactions of specific VOCs with OH and ozone to the Atmospheric Oxidative Capacity in Section 3.3.

*Line 424 – The increase in OH in Hong Kong appears very strong as well.*

Author's reply: We selected four city sites (Beijing, Shanghai, Guangzhou, Chengdu sites) to discuss the changes in OH radical. As the location of Hong Kong site is close to the Guangzhou site, we deleted the description for the Hong Kong sites in the main text.

*Line 473 – Does the model really have NH3 + OH as a significant sink of OH? If the authors are referring to its impact on SIA and thus HO2 uptake, this is not clear.*

Author's reply: Our model includes the reaction between $NH_3$ and OH, but it is not a significant sink of OH. The increase in OH radical represented here is mainly due to the less consumption by reduced CO concentrations. We changed the statement as follows (Line 284-293 in Text):

"When accounting for the additional reduction in other anthropogenic emissions ($NH_3$, $SO_2$, and CO) (*TOTAL* case), the mixing ratio of the OH radical is positively modified, relative to the results in the combined case (*N+A* case). As shown in Fig. S4a, the mixing ratio of the OH radical is enhanced in the PRD and SCB regions (by up to 22%). This increase is due to the lowered consumption of the OH radical by the reduced concentration of carbon monoxide (CO) (Fig. S5a), due to its reduced emissions (Fig. S1d). For the $HO_2$ radicals, the additional reduction in the other emissions also contributes to a larger mixing ratio, with a pronounced increase in southern China (by up to 18%; Fig. S4c). This increase in the $HO_2$ radical mixing ratio is due to the increase in the oxidation of the VOCs by the OH radical and the reduced aerosol uptake of $HO_2$ associated with the decrease in the aerosol load".

*Paragraph starting on line 567 – This discussion is again very lengthy. The figure appears to show that the most important message is that NOx reductions in July alone result in ozone decreases in several major cities (Beijing, Shanghai, Hong Kong?) while adding in AVOC reductions causes the cities to also see an ozone decrease.*

Author's reply: To structure our description, we discussed the ozone changes due to emissions reduction in winter and summer conditions separately. We also summarized the specific ozone changes at four urban sites in Table 1 (Table 2 in Text). and the relevant statement of ozone changes is shown below (Section 3.2.2 in Text).

"*Winter conditions.* In January, the 50% reduction in $NO_x$ emission enhanced the surface ozone concentrations, with the largest increase derived in the YRD and PRD regions (15-20% (8-10 ppbv); Fig. 6a). During wintertime, a large part of China is under

a VOC-sensitive regime. Therefore, the reduced titration of ozone by reduced NO (Fig. S13a; Fig. S2c) favors an increase in the ozone concentration. If AVOCs emissions are reduced by 50%, the surface ozone is reduced by 4-10% (2.0 to 8.0 ppbv; Fig. 6b) in the southern part of China. This ozone decrease is associated with the reduced concentration of $HO_x$ radicals and hence a reduction in the ozone production by the $HO_2 + NO$ reaction (Fig. S14a).

In the combined emission reduction case, the ozone response in VOC-limited areas follows the positive changes found in the $NO_x$-reduction case, with an ozone increase of 3.0-7.5 ppbv (4-9%) in North China and in some urban regions in South China (Fig. 6c). Simultaneously, a slight ozone decrease is derived over the southern coast of China (2.0-4.5 ppbv; 5-8%). In these areas, the ozone sensitivity is under the control of the $NO_x$. The ozone decrease is dominant by the negative ozone response to the AVOCs emissions reduction. With further emission reduction of the other species, an ozone increase (3-5 ppbv (4-6%); Fig. S5g) relative to the combined case is calculated in the southern part of China.

*Summer condition.* In July, under the reduction in the $NO_x$ emissions, an increase in the surface ozone concentration of up to 10 ppbv (17%) is calculated in the urbanized regions of NCP, YRD, and PRD (Fig. 6d). These areas are typically located in VOC-limited areas (Fig. 5); thus, the ozone increase is explained by the reduced ozone titration (Fig. S13b). At the same time, in $NO_x$-limited areas, the calculated surface ozone concentration is reduced by 2 to 8 ppbv (3-10%), as a result of reduced photochemical formation under lower $NO_x$ concentrations. With the reduction of AVOCs emissions, the surface ozone concentration decreases by up to 8.0-12.0 ppbv (8-20%; Fig. 6e) in whole areas of China. A spatial shift in the ozone decrease, from the southern regions in winter to the northern regions in summer occurs under this condition; this change is consistent with the spatial distribution of the reduction in the mixing ratio of the $HO_2$ radical, which contributes to the ozone production by its reaction with nitric oxide (Fig. S14b).

When combining the 50% reduction in the $NO_x$ and AVOCs emissions, the surface ozone concentration decreases by up to 12 ppbv (15%; Fig. 6f) in $NO_x$-sensitive areas. In VOC-sensitive areas, the surface ozone concentration also decreases, as the increase of ozone associated with the positive impact of the reduction in $NO_x$ emissions is smaller than the negative effect resulting from the reduction in the AVOCs emissions. This is explained by the fact that the loss of ozone due to the reduced $NO_x$ level is rapidly compensated by the photochemically ozone formation processes, since the ozone production rate is accelerated by the high temperature and photolysis rate during summertime (T. Wang et al., 2022). One exception can be found at the Guangzhou site, where ozone slightly increases by 0.5 ppbv (Fig. S15), which can be explained by the increasingly important role of naturally emitted BVOCs species in the oxidation processes when anthropogenic emissions are reduced (see Sec. 3.3). When the emission reduction is applied to all species under consideration, the ozone changes (Fig. S5h) relative to the combined case are smaller than the changes in winter, due to a

consistently smaller reduction in aerosol concentrations (see Sec. 3.2.3)."

Table 1. Ozone changes due to reduction in emissions in urban sites (in percentage)

| Location | Sites name | Ozone changes in winter condition (Mean ± SD) | | | |
|---|---|---|---|---|---|
| | | NOx[a] | AVOCs[b] | N+A[c] | TOTAL[d] |
| North | Beijing | 25.0 ± 25.2[e] | -2.5 ± 1.3 | 22.0 ± 32.8 | 20.0 ± 19.5 |
| East | Shanghai | 33.2 ± 35.3 | -18.2 ± 13.5 | 21.8 ± 20.5 | 22.7 ± 18.8 |
| South | Guangzhou | 21.4 ± 22.6 | -17.1 ± 11.2 | 7.1 ± 3.2 | 10.0 ± 3.5 |
| West | Chengdu | 21.3 ± 23.8 | -9.4 ± 8.5 | 14.1 ± 8.3 | 20.3 ± 13.5 |
| Location | Sites name | Ozone changes in summer condition (Mean ± SD) | | | |
| | | NOx | AVOCs | N+A | TOTAL |
| North | Beijing | 6.4 ± 3.8 | -21.8 ± 19.2 | -5.5 ± 4.2 | -7.3 ± 5.0 |
| East | Shanghai | 17.1 ± 12.8 | -22.9 ± 20.8 | -2.9 ± 2.1 | -2.6 ± 1.5 |
| South | Guangzhou | 15.0 ± 13.1 | -14.5 ± 13.5 | 1.3 ± 1.0 | 1.3 ± 0.9 |
| West | Chengdu | 5.5 ± 4.5 | -14.5 ± 10.2 | -5.5 ± 2.0 | -4.5 ± 1.9 |

a-d. Sensitivity cases with a 50% reduction in $NO_x$ emissions (NOx), AVOCs emissions (AVOCs), NOx and AVOCs (N+A), and other species (NOx, AVOCs, CO, $NH_3$, $SO_2$) under consideration (TOTAL).

e. Values are displayed in the average ozone changes during daytime (06:00-19:00) in percentage with the standard deviation as the error bar.  (ozone changes = (case value -base-line case) *100).

*Line 599 – How do reduced AVOCs impact nitrate, sulfate, and ammonium?  It is not clear from this sentence.*

Author's reply:  The aerosol decrease due to the reduction in AVOCs emissions is attributed to the decrease in SOA, with minor impact on SIA. We changed the statement as follows (Line 502-505 in Text):

"With a 50% reduction of AVOCs emissions, the changes in the aerosol concentration are smaller than with the 50% reduction in $NO_x$ emissions, with a decrease of less than 4% (5 µg m$^{-3}$; Fig. 7b), which predominantly results from the reduction in SOA (Fig. S18a).

*Line 628 – How much does photolysis increase in your model with reduced aerosol?*

Author's reply: The photolysis increases by about 5-20% in winter and 3-10% in summer, as shown in Figure 3 (Fig. S20 in Text). We added some description of aerosol-related increase in photolysis and its potential impact ozone formation in the text shown below (Line 534-541 in Text).

"This decrease in the aerosol burden weakens the aerosol extinction effect and therefore enhances the photochemical formation rate of radicals and ozone. As shown in Fig. S20 a-d, the photolysis rate increases (by 5-20%) in southern and central China during winter due to the aerosol decrease induced by the emission reductions. The highest increase in photolysis rates results from the joint emission reduction in $NO_x$ and AVOCs (Fig. S20c). The increase of the photolysis rates in summer is not as distinct as the increase during wintertime due to the more limited reduction of the aerosol burden during summer (Fig. S20e-h)."

[Figure]

**Figure 3.** Changes in the total photolysis rate (a-d) [Unit: s$^{-1}$] due to the emission reduction in NOx (a,e), AVOCs (b,f), NOx and AVOCs (N+A)(c, g) relative to the BASE case and TOTAL case relative to N+A cases (d, h) in January (a-d) and July (e-h) of 2018.

*Figure 13 – What is in 'Other' that is impacted in your 'TOTAL' case? This category is a surprisingly large fraction of model PM2.5 and thus deserves more discussion. Overall, Fig. 13 contains a lot of information but is barely discussed.*

Author's reply: The large decrease in the aerosol load for the "TOTAL" case is due to the reduction in the sulfate and ammoniate particulates, as the emissions of $SO_2$ and $NH_3$ are reduced in this case. The relevant description is shown below (Line 510-514 in Text):

"With a further reduction in other emissions, the decrease in the concentration of aerosol is deeply enhanced; this is the case for the concentration of $NH_4^+$ (919a), $SO_4^{2-}$ (Fig. S19b), and $NO_3^-$ particles (Fig. S19c). The concentration of the gas-phase precursors, $NH_3$ and $SO_2$, is considerably reduced, which affects the process of acid replacement (Meng et al., 2022) and hence the level of $NO_3^-$."

*Line 645 – Why have a schematic for reduction in $NO_x$ emissions, but not AVOC emissions, and the combination of the two?*

Author's reply: We changed the schematic and show it below.

[Figure]

Figure 4. Schematics show the responses of oxidative processes, associated with ozone formation, to the reduction in primary emissions of $NO_x$ and AVOCs in urban areas (VOC-limited) in winter and summer. Arrows besides the chemicals represent the changes associated with the reduction in emission. (decrease trend shown in blue; increase trend shown in red) Blue and red arrows closing to $O_3$ represent the positive and negative contributions to the ozone formations. *AOC*, *P(O₃)*, and *D(O₃)* are the abbreviations of the Atmospheric Oxidative Capacity, production of ozone, and

destruction of ozone. Bar figure shows the ranges of ozone changes in whole of China (black bar), in non-urban areas (white part in the bar), and in urban areas (colored part in the bar) in three emissions cases (NOx, AVOCs, and N+A represent the case with emissions reduction in $NO_x$, Anthropogenic VOCs (AVOCs), and the combined $NO_x$ and AVOCs emissions, respectively) relative to BASE cases in winter and summer.

*Section 3.4.1 – Again, I am not sure what the main message is from this lengthy section.*

Author's reply:  We largely condensed the paragraph and kept the discussion about the changes in daytime *AOC* in response to emission reduction, as it gives us insights on the contributions from the reaction between specific VOCs with OH and $O_3$.

*Line 755 – What is the result of the increased ozonolysis?  Do we get more OVOCs that impact daytime air quality? This is said later but is not clear here.*

Author's reply:  The changes in atmospheric oxidative capacity due to the increased ozonolysis are shown in Figure 5a-b. The concentration of OVOCs increases when $NO_x$ emissions are reduced as shown in Figure 5c-d. The relevant statement in the main text is also shown below (Line 571-573 in Text).

"During nighttime (20:00 to 05:00 LST), the reduction in $NO_x$ emissions is responsible for an increase in *AOC* by up to 50% (Fig. S21a). A contribution to this increase is provided by the alkene's ozonolysis, since the concentration of ozone (Fig. 6 a) and of alkenes is enhanced (Fig. S8c)."

[Figure]

**Figure 5.** Changes in atmospheric oxidizing capacity (*AOC*, Unit: $10^6$ molec. cm$^{-3}$ s$^{-1}$) due to the reaction between alkenes and ozone (ozonolysis) (a, b) and the reaction between OH and OVOCs (c, d) in response to the $NO_x$ emissions case relative to the

BASE case in January (a, c) and July (b, d) of 2018.

*Line 771 – Are these primary OVOCs like methanol or ethanol? Or secondary species like HCHO and acetaldehyde? If secondary, then what are their main precursors? Which 'unsaturated OVOCs' should be targeted?*

Author's reply: In the calculation of atmospheric oxidative capacity (*AOC*), the reaction related OVOCs includes all OVOCs species (primary and secondary). Considering the contributions of VOCs-related reactions to *AOC* increase and the increases in OVOCs species associated with $NO_x$ emissions reduction, we suggest the reduction in the emissions of alkenes, aromatics, and unsaturated OVOCs, especially methanol and ethanol. The relevant statement is shown below (Line 687-689 in Text).

"With the known contribution of the VOCs-related reactions to the *AOC*, the reduction in the emissions of alkenes, aromatics, and unsaturated OVOCs, especially the methanol and ethanol, should be a priority."

*Summary – Again, a greater focus on policy-relevant insights would be helpful as there are opposite effects on average compared to in the major cities.*

Author's reply: we added some policy implications for the ozone mitigation in summary as shown it below.

*"Paths to mitigation.* We conclude this paper by highlighting a few chemical paths that should be considered when designing a mitigation policy for a reduction of ozone in the urban areas of China. Figure 10 presents a schematic description of the chemical mechanisms involved in the chemical production of atmospheric ozone and highlights how different reaction paths tend to change the ozone abundance in response to a reduction in $NO_x$ and anthropogenic VOC (AVOCs) emissions. This graph shows that a reduction in $NO_x$ emissions tends to increase the ozone concentration by (1) reducing the rate of the $NO + O_3$ reaction (ozone titration); (2) by increasing the rate of the $HO_2 + NO$ reaction due to an increase in the $HO_2$ level associated with the reduced uptake of this radical by a lowered aerosol load; (3) by an increase in the atmospheric oxidizing capacity (*AOC*) through OH- and ozone-related reactions. The graph also shows that a decrease in AVOCs emissions tends (1) to reduce the level of the $HO_x$ radical and hence the ozone production by the $HO_2 + NO$ reaction; (2) to enhance the level of $HO_x$ due to the reduced aerosol uptake and (3) to reduce the *AOC* with a negative effect on the ozone concentration. The relative importance of these different chemical mechanisms varies with location and environmental conditions.

We conclude that, in winter when the background ozone concentration is low, the reduction of $NO_x$ emissions tends to increase the level of near-surface ozone, while the reduction in AVOC emissions has the opposite effect. This conclusion applies both in rural and in urban areas. A combined reduction in the emissions of these two primary

pollutants tends to decrease the level of ozone in rural areas, but to increase ozone in urban areas. Thus, in urban areas during winter, an effective approach to reduce the surface ozone concentration is through a strong limitation in the emissions of volatile organic compounds.

In summer when the ozone level is generally high, the reduction of $NO_x$ emissions is an effective action to reduce the ozone concentration in rural areas, but this measure is counterproductive in the NOx-saturated urban areas where ozone is controlled by VOCs. In fact, in urban areas during this season, the mechanisms involved in ozone mitigation are complex. For example, when $NO_x$ emissions are reduced, the atmospheric OH concentration is enhanced because of its reduced destruction by $NO_2$. Following this increase in the OH concentration, an increase in the level of OVOCs, whose photolysis is an important source of $HO_x$ radicals, also leads to accelerated ozone production and further amplifies the oxidation of VOCs. In addition, the increase in *AOC*, linked to the reaction of OH and ozone with alkenes and the reactions of OH with OVOCs also contribute to an increase in the ozone production. Further, the reduction in the aerosol load resulting from a reduction in the emissions of aerosol precursors promotes the ozone formation by decreasing the aerosol extinction and by reducing the uptake of $HO_2$. If combined with a 50% reduction in AVOCs, the increase in OVOCs and *AOC*, due to reduced $NO_x$ emissions, can be offset. However, the aerosol-related promotion of the level of OH and $HO_2$ radicals can be enhanced, highlighting the complexity of summertime ozone mitigation in urban areas.

Table 2 provides quantitative information on the response of ozone at different urban locations for January and July. In urban areas, the reduction in the level of surface ozone requires a reduction in the emissions of anthropogenic VOCs. However, for practical reasons, a 50% reduction in AVOCs emissions, as assumed in our study, is difficult to implement over a short period of time. With the known contribution of the VOCs-related reactions to the *AOC*, the reduction in the emissions of alkenes, aromatics, and unsaturated OVOCs, especially the aldehydes and alcohols, should be a priority. The development of efficient mitigation strategies based on the reduction of AVOCs emissions requires, however, more detailed investigations on the reactivity of individual VOCs and on their potential impact on the ozone formation."

*Line 795 – What about reduced loss of OH to OH + NO2 which increases the ability to oxidize VOCs?*

Author's reply: We believe the increase of OH due to the reduced loss of OH reacted with $NO_2$ is an important pathway to the increased ozone formation. We provide a comprehensive statement about how ozone increases in VOCs-limited aeras (Figure 4) and show it below (Line 644-651 in Text).

"Figure 10 presents a schematic description of the chemical mechanisms involved in the chemical production of atmospheric ozone and highlights how different reaction

paths tend to change the ozone abundance in response to a reduction in $NO_x$ and anthropogenic VOC (AVOCs) emissions. This graph shows that a reduction in $NO_x$ emissions tends to increase the ozone concentration by (1) reducing the rate of the NO + $O_3$ reaction (ozone titration); (2) by increasing the rate of the $HO_2$ + NO reaction due to an increase in the $HO_2$ level associated with the reduced uptake of this radical by a lowered aerosol load; (3) by an increase in the atmospheric oxidizing capacity (*AOC*) through OH- and ozone-related reactions."

*Line 830 – The reason for the greater joint impact needs to be explained.*

Author's reply: we added some explanations for the larger decrease in aerosol in the joint case with the sentence shown below (Line 505-508):

"With a joint reduction in $NO_x$ and AVOCs (Fig. 7c), the aerosol decrease is larger than the separated effect of the individual emissions decrease, as the increase in the concentration of SOA resulting from the reduced $NO_x$ emissions is compensated by the reduced AVOCs emissions."

*Line 853 – Refrain from discussing 'slight' changes to focus on the major findings.*

Author's reply: We deleted the discussion regarding slight changes.

*Line 869 – Here the authors state that their goal is to help develop a strategy for metropolitan areas. If this is the goal of the paper, the authors should consider a greater focus on the impacts on cities (bar chart figures such as Fig. 4).*

Author's reply: We agree with this statement. We focused our discussion of ozone changes and relevant policy implication at four city sites, as shown in the schematics (Figure 4) of our study.

*Line 869 – The authors already specifically call out categories of VOCs (alkenes, aromatics etc). Could the authors better describe what they mean by 'more detailed investigations' here?*

Author's reply: We elaborated more on the detailed investigation as (Line 683-685 in Text)

"The development of efficient mitigation strategies based on the reduction of AVOCs emissions requires, however, more detailed investigations on the reactivity of individual VOCs and on their potential impact on the ozone formation"

*Code and data availability: This does not include the modifications made to WRF-Chem described in Dai et al., 2023 and used here.*

Author's reply: We are willing to share our code upon asking, and we emphasize this in the code availability part.

---

## Author Comment (AC2)

**Response to Reviewers' Comments**

**Dear Editor and Reviewers,**

Thank you very much for your efforts in handling and evaluating our submission.
The review comments are very helpful for improving the original manuscript. We have carefully considered and tried to address all of these comments in the revised manuscript. Below are the detailed point-by-point responses to the review comments. For clarity, the reviewer's comments are listed below in *black italics*, while our responses and changes in the manuscript are highlighted in blue and red, respectively. We look forward to receiving a further evaluation of our work.

Best regards,
Guy Brasseur and co-authors

**Summary**

The authors perform four sets of model simulations over China: a base case, a 50% NOx emission reduction, a 50% AVOC emission reduction, and a combined 50% NOx and 50% AVOC emission reduction. The science presented is largely sound. However, the paper currently reads as a lengthy report rather than a scientific manuscript. There is extensive repetition of conclusions in different sections of the paper, and a lot of the more impactful conclusions get lost in the details. I recommend that the authors restructure the paper around their main conclusions and science questions, rather than organizing by metric as is done currently. I also suggest that many of the minor details presented in the main text be moved to the supplement so that only text in support of the main conclusions of the paper is presented in the main text.

**Major comments**

*As is currently written, a lot of the main conclusions and the primary storyline get lost in the details presented. In addition, a lot of conclusions get repeated in different sections. For example, a lot of the same conclusions are drawn when assessing ozone production regime (Section 3.1), odd oxygen production and destruction (Section 3.2.2), and ozone concentrations (Section 3.3.2). Uniting these sections will reduce the overall manuscript length, help highlight whether these different lines of analysis lead to consistent conclusions and help to bring the overall conclusions of the study to the forefront.*

*I wonder if the authors could comment on the relevance of broad NOx and AVOC reductions in China, in contrast to reductions that vary by sector or by region. For example, are transportation sector reductions more/less likely than stationary emissions, and what might this imply for the chemistry discussed? In addition, a lot of the analysis presented highlighted changes in radical cycling related to changes in OVOC emissions. Do we expect AVOC emission reductions to be consistent across classes of VOCs, or could the effectiveness of emission reductions of OVOCs vs hydrocarbons differ? And, similarly, do we expect consistent emission reductions for VOCs with higher and lower HCHO yields? How might this impact the conclusions presented here?*

Author's reply: Thank you very much for your comments that are very helpful. Based on your comments, we significantly condensed our paper, with a shortened description of the text on our model results and are focusing on insights about how to mitigate ozone increase in urban China. For the metrics related to photochemical activity, we limited our analysis to the Atmospheric Oxidative Capacity (AOC) in the main text and moved other calculated metrics to the Supplement. The structure of this paper is as follows.

"This paper is structured as follows. Section 2 introduces the setups of the model system and describes the simulations performed for specified reductions in the emissions of

primary pollutants. In Section 3, we first analyzed the response in the near-surface concentration of ozone precursors and intermediate to primary emission reductions. Then, we also discuss the changes in the ozone formation regime. Further, we derive the associated changes in ozone, and aerosols to emission reductions. Finally, we describe the sensitivity of the atmospheric oxidative capacity ($AOC$) to the reduction in emissions. A summary and implication for policy making of our study is provided in Sec. 4."

Regarding policy to mitigate ozone, we highlight that the reduction in emissions should be implemented by regions and by the type of environment, with different strategies for the south and in the north of China and for the urban versus non-urban areas. We suggest that the reduction in $NO_x$ emissions be coordinated with the reduction in AVOCs emissions, especially with reduction of alkenes, aromatics and unsaturated OVOC emissions, including methanol or ethanol. This conclusion is based on the contribution of these different species to the daytime oxidative capacity of atmosphere and to the secondary formation of OVOCs. The summary of our policy implication [also shown in Table 1 and Figure 1 (Table 2 and Figure 10 in Text)] is shown below.

"*Paths to mitigation.* We conclude this paper by highlighting a few chemical paths that should be considered when designing a mitigation policy for a reduction of ozone in the urban areas of China. Figure 10 presents a schematic description of the chemical mechanisms involved in the chemical production of atmospheric ozone and highlights how different reaction paths tend to change the ozone abundance in response to a reduction in $NO_x$ and anthropogenic VOC (AVOCs) emissions. This graph shows that a reduction in $NO_x$ emissions tends to increase the ozone concentration by (1) reducing the rate of the $NO + O_3$ reaction (ozone titration); (2) by increasing the rate of the $HO_2 + NO$ reaction due to an increase in the $HO_2$ level associated with the reduced uptake of this radical by a lowered aerosol load; (3) by an increase in the atmospheric oxidizing capacity ($AOC$) through OH- and ozone-related reactions. The graph also shows that a decrease in AVOCs emissions tends (1) to reduce the level of the $HO_x$ radical and hence the ozone production by the $HO_2 + NO$ reaction; (2) to enhance the level of $HO_x$ due to the reduced aerosol uptake and (3) to reduce the $AOC$ with a negative effect on the ozone concentration. The relative importance of these different chemical mechanisms varies with location and environmental conditions.

We conclude that, in winter when the background ozone concentration is low, the reduction of $NO_x$ emissions tends to increase the level of near-surface ozone, while the reduction in AVOC emissions has the opposite effect. This conclusion applies both in rural and in urban areas. A combined reduction in the emissions of these two primary pollutants tends to decrease the level of ozone in rural areas, but to increase ozone in urban areas. Thus, in urban areas during winter, an effective approach to reduce the surface ozone concentration is through a strong limitation in the emissions of volatile organic compounds.

In summer when the ozone level is generally high, the reduction of $NO_x$ emissions is an effective action to reduce the ozone concentration in rural areas, but this measure is counterproductive in the NOx-saturated urban areas where ozone is controlled by VOCs. In fact, in urban areas during this season, the mechanisms involved in ozone mitigation are complex. For example, when $NO_x$ emissions are reduced, the atmospheric OH concentration is enhanced because of its reduced destruction by $NO_2$. Following this increase in the OH concentration, an increase in the level of OVOCs, whose photolysis is an important source of $HO_x$ radicals, also leads to accelerated ozone production and further amplifies the oxidation of VOCs. In addition, the increase in *AOC*, linked to the reaction of OH and ozone with alkenes and the reactions of OH with OVOCs also contribute to an increase in the ozone production. Further, the reduction in the aerosol load resulting from a reduction in the emissions of aerosol precursors promotes the ozone formation by decreasing the aerosol extinction and by reducing the uptake of $HO_2$. If combined with a 50% reduction in AVOCs, the increase in OVOCs and *AOC*, due to reduced $NO_x$ emissions, can be offset. However, the aerosol-related promotion of the level of OH and $HO_2$ radicals can be enhanced, highlighting the complexity of summertime ozone mitigation in urban areas.

Table 2 provides quantitative information on the response of ozone at different urban locations for January and July. In urban areas, the reduction in the level of surface ozone requires a reduction in the emissions of anthropogenic VOCs. However, for practical reasons, a 50% reduction in AVOCs emissions, as assumed in our study, is difficult to implement over a short period of time. With the known contribution of the VOCs-related reactions to the *AOC*, the reduction in the emissions of alkenes, aromatics, and unsaturated OVOCs, especially the aldehydes and alcohols, should be a priority. The development of efficient mitigation strategies based on the reduction of AVOCs emissions requires, however, more detailed investigations on the reactivity of individual VOCs and on their potential impact on the ozone formation."

Table 1. Ozone changes due to reduction in emissions in urban sites (in percentage)

| Location | Sites name | Ozone changes in winter condition (Mean ± SD) | | | |
|---|---|---|---|---|---|
| | | NOx[a] | AVOCs[b] | N+A[c] | TOTAL[d] |
| North | Beijing | 25.0 ± 25.2[e] | -2.5 ± 1.3 | 22.0 ± 32.8 | 20.0 ± 19.5 |
| East | Shanghai | 33.2 ± 35.3 | -18.2 ± 13.5 | 21.8 ± 20.5 | 22.7 ± 18.8 |
| South | Guangzhou | 21.4 ± 22.6 | -17.1 ± 11.2 | 7.1 ± 3.2 | 10.0 ± 3.5 |
| West | Chengdu | 21.3 ± 23.8 | -9.4 ± 8.5 | 14.1 ± 8.3 | 20.3 ± 13.5 |
| Location | Sites name | Ozone changes in summer condition (Mean ± SD) | | | |
| | | NOx | AVOCs | N+A | TOTAL |
| North | Beijing | 6.4 ± 3.8 | -21.8 ± 19.2 | -5.5 ± 4.2 | -7.3 ± 5.0 |
| East | Shanghai | 17.1 ± 12.8 | -22.9 ± 20.8 | -2.9 ± 2.1 | -2.6 ± 1.5 |
| South | Guangzhou | 15.0 ± 13.1 | -14.5 ± 13.5 | 1.3 ± 1.0 | 1.3 ± 0.9 |
| West | Chengdu | 5.5 ± 4.5 | -14.5 ± 10.2 | -5.5 ± 2.0 | -4.5 ± 1.9 |

a-d. Sensitivity cases with a 50% reduction in $NO_x$ emissions (NOx), AVOCs emissions (AVOCs), NOx and AVOCs (N+A), and other species (NOx, AVOCs, CO, $NH_3$, $SO_2$) under consideration (TOTAL).

e. Values are displayed in the average ozone changes during daytime (06:00-19:00) in percentage with the standard deviation as the error bar. (ozone changes = (case value -base-line case) *100).

[Figure]

Figure 1. Schematics show the responses of oxidative processes, associated with ozone formation, to the reduction in primary emissions of $NO_x$ and AVOCs in urban areas (VOC-limited) in winter and summer. Arrows besides the chemicals represent the changes associated with the reduction in emission. (decrease trend shown in blue; increase trend shown in red) Blue and red arrows closing to $O_3$ represent the positive and negative contributions to the ozone formations. *AOC*, $P(O_3)$, and $D(O_3)$ are the abbreviations of the Atmospheric Oxidative Capacity, production of ozone, and destruction of ozone. Bar figure shows the ranges of ozone changes in whole of China (black bar), in non-urban areas (white part in the bar), and in urban areas (colored part in the bar) in three emissions cases ($NO_x$, AVOCs, and N+A represent the case with emission reduction in $NO_x$, Anthropogenic VOCs (AVOCs), and the combined $NO_x$ and AVOCs emissions, respectively) relative to BASE cases in winter and summer conditions.

**Minor comments**

*Line 110: typo (nitration vs titration)*

Author's reply: Revised.

*Line 236: Zhang et al, 2009 get these numbers from Tonnesen and Dennis, 2000, so Tonnesen and Dennis, 2000 should be cited here as the original citation.*

Author's reply: Changed.

---

## Referee Report (RR1)

**General Comments**

The authors have chosen to look at the impact of emissions changes on January and July ozone in China. The manuscript is improved over the first draft, however there are still some organizational changes that could be made to improve the impact of this paper. Specifically, shifting the discussion to first focus on absolute ozone changes, and then focus on the drivers of those changes, could help avoid inconsistencies. Finally, the way the paper is written, it is not clear why the 'TOTAL' case belongs in the supplement instead of the main paper. The paper would be more impactful as well if the abstract clearly spelled out the big picture relationship between their emission scenarios, and the impact on both gas and aerosol-phase. Currently the abstract only discusses ozone but none of the complexity related to aerosol impacts for example on HO2 uptake.

**Specific Comments**

Abstract – The authors make no mention of their conclusions related to aerosol changes.

Line 258 – The authors state "A distinct increase in the surface mixing ratio of HO2 radical is derived in southern China (by up to 5 pptv or 60%; Fig. 1b). This enhancement is related to the increased mixing ratio of the OH radical found in urban areas, resulting in enhanced HO2 levels via VOCs oxidation, and a reduced HO2 loss via the aerosol uptake, as the aerosol load is reduced (see Sect. 3.2.3) (Song et al., 2021)." However the increased HO2 appears to be present almost everywhere, while the increase in OH is highly localized to urban areas as the authors state. The authors should be more clear that the broader increase in HO2 must be due to the reduced HO2 loss via aerosol uptake. If the authors tracked production from that chemical pathway, they could show a map of that rate decrease which could help. The authors could also consider a plot of the decrease in PM2.5.

Figure 1&2 – I would suggest adding the 'TOTAL' case to Figure 1 and Figure 2. Figure S4b appears to be mis-labeled as January instead of July.

Line 304 – What do you mean by "These changes are affected by meteorological parameters including the temperature, the water vapor abundance, and the solar radiation intensity, which affect the oxidative processes (Dai et al., 2023)." Is meteorology not the same in both simulations?

Line 342 – It looks like the abundance of OVOCs is reduced in "all" regions, not "most" regions.

Line 347 – Can you explain why this is? "the decrease is the most pronounced in the concentration of ketones" Why does the concentration of alcohols for example not seem to change at all? Is the model budget of alcohols really dominated by BVOCs, and if so, could that really be correct?

Line 376 – Is it more effective OVOC production? Or biogenic emission of OVOCs? Maybe a budget of OVOCs (AVOC vs. BVOC vs. secondary production) would help?

Line 411 – There is still a lot of VOC-limited area. Instead of "tend to be converted", maybe say x% of VOC-limited is converted to transition or NOx-limited?

Line 414 – Just confirming that your $HO_2$ uptake reaction does not produce $H_2O_2$? Does it produce $H_2O$?

Line 420 – Against suggest pulling the 'TOTAL' case in to the main text.

Line 431 – Can we learn something from Guangzhou? Does Guangzhou have differences in emissions compared to the other cities that would explain why it remains VOC-limited? In the N+A and TOTAL cases, this applies also to Shanghai? What is the difference compared to Beijing and Chengdu?

Figure 6 – Shouldn't the unit be (ppbv) not (pptv)?

Table 2 shows that in winter, $NO_x$ reduction results in ozone increase in all cities, and AVOC reduction results in ozone decrease in all cities. The N+A and TOTAL cases result in ozone increases in all cities. In summer, NOx

reduction results in ozone increases in all cities while AVOC reduction results in decreases in all cities. In the N+A and TOTAL cases, ozone decreases in Beijing, Shanghai, and Chengdu, but not in Guangzhou. According to Figure 5, in July, in the N+A and TOTAL cases, Guangzhou and Shanghai remain VOC-limited while Beijing and Chengdu shift to transitional conditions. Given that you get a different picture from Table 2 vs. Figure 5 (in Table 2, Guangzhou stands out) but in Figure 5, Guangzhou and Shanghai are different from Beijing and Chengdu, it might help to start with the ozone changes in Table 2, and use your other analysis to explain those changes, rather than starting with radical changes and NOx vs. VOC-limited changes.

Figure 7 – I think it would be better if 7b was on the same scale as 7a and 7c.

Line 499 – The meaning of this is unclear "followed by effect of $NO_{4+}$".

Line 515 – Cite Dai et al., 2023 here for this model bias evaluation?

Line 550 – Please add some discussion of the model HO2 uptake parameterization and uncertainties in the strength of this uptake (for example, is gamma 0.2 or 0.1)? Previous studies have reduced this gamma to better fit observations (e.g., Yang et al., 2023).

Line 555 – Can you better describe the calculation of AOC? Is there an equation you can add here?

Line 557 – There is nothing in the discussion below to support this statement: "This parameter allows us to characterize the formation process of $O_3$ and can be used as an indicator to design mitigation policies for reducing ozone pollution." I think this comes in better in the conclusions where you describe the relative importance of different VOCs to AOC. How is this different/better than the use of OH reactivity? The conclusions mention that AOC helps you to pick out "alkenes, aromatics, and unsaturated OVOCs, especially methanol and ethanol." It would help if the identification of those VOCs were discussed in Section 3.3 and not solely placed in the conclusions.

Line 640 – Better to name the specific cities and instead of 'slight' give the actual increase.

Line 651 – Does ozonolysis really have a net impact of increasing ozone levels? Just need clarification here on the suggestion that the net effect is positive.

Line 653 – Do you mean enhance the level of OH? Otherwise this is a nice schematic and helpful description.

Line 704 – The authors state: "The modified code in the WRF-Chem model is available upon request to the corresponding author." Best practice now seems to be to put modified code on Zenodo.

**References**

Yang, L. H., Jacob, D. J., Colombi, N. K., Zhai, S., Bates, K. H., Shah, V., Beaudry, E., Yantosca, R. M., Lin, H., Brewer, J. F., Chong, H., Travis, K. R., Crawford, J. H., Lamsal, L. N., Koo, J.-H., and Kim, J.: Tropospheric $NO_2$ vertical profiles over South Korea and their relation to oxidant chemistry: implications for geostationary satellite retrievals and the observation of $NO_2$ diurnal variation from space, Atmos. Chem. Phys., 23, 2465–2481, https://doi.org/10.5194/acp-23-2465-2023, 2023.

---

## Author Response (AR2)

**Response to Reviewers' Comments**

**Dear Editor and Reviewers,**

Thank you very much for your efforts in handling and evaluating our submission.
The review comments are very helpful for improving the original manuscript. We have carefully considered them and tried to address all of these comments in the revised version of the manuscript. Attached are the detailed point-by-point responses to the review comments. For clarity, the reviewer's comments are listed below in *black italics*, while our responses and changes in the manuscript are highlighted in blue and red, respectively.
We look forward to receiving a further evaluation of our work.

Best regards,
Guy Brasseur and co-authors

**General Comments**

*The authors have chosen to look at the impact of emissions changes on January and July ozone in China. The manuscript is improved over the first draft, however there are still some organizational changes that could be made to improve the impact of this paper. Specifically, shifting the discussion to first focus on absolute ozone changes, and then focus on the drivers of those changes, could help avoid inconsistencies. Finally, the way the paper is written, it is not clear why the 'TOTAL' case belongs in the supplement instead of the main paper. The paper would be more impactful as well if the abstract clearly spelled out the big picture relationship between their emission scenarios, and the impact on both gas and aerosol-phase. Currently the abstract only discusses ozone but none of the complexity related to aerosol impacts, for example on HO2 uptake.*

Author's reply: Thanks for your insightful suggestions. (1) We have improved the organization of our manuscripts with the first focus on the discussion of absolute ozone changes, followed by the explanation of the drivers for the ozone changes. The structure of our paper is shown as follows (See Line 159 to 164 in Text).

"In Section 3, we analyze the response of near-surface concentration of ozone to the specified emission reductions. Further, we determine the drivers responsible for the resulting ozone changes; these include changes in the concentrations of ozone precursors, of the intermediates including the oxidized VOCs (OVOCs) and in the level of secondary aerosols. We also discuss the changes to be expected in the ozone formation regimes. Finally, we describe the sensitivity of the atmospheric oxidative capacity (*AOC*) to the reduction in the emissions."

(2) The results of TOTAL case have been moved from supplementary materials to main text. (3) The abstract has also been improved with comprehensive description of the impact of emission reduction to ozone changes with both gas and aerosol-phase reactions, and shown it below:

"Despite substantial reductions in anthropogenic emissions, ozone (O3) pollution remains a severe environmental problem in urban areas of China. The reduction in the emission of pollutants affects formation of ozone through the changes in concentrations of O3 precursors and intermediates species as well as in the oxidation capacity of the atmosphere. However, the underlying mechanisms driving O3 changes are still not fully understood. Here, we employ a regional chemical transport model to quantify the changes in the formation of ozone as well as other secondary pollutants to a specified emission reduction (50%) for winter and summer conditions (January and July 2018). Our results indicate that, in winter, a 50% decrease in nitrogen oxide (NOx) emissions leads to an increase in surface O3 concentrations of 15%−33% on average across China. In summer, the concentration of O3 increases by up to 17% in the areas limited by the level volatile organic compounds (VOCs), while it decreases by 3%−12% in NOx-limited areas. The increase in the ozone concentration is associated with a reduced NOx-titration effect and higher levels of hydroxyl (OH) due to a reduced loss from

reactions with nitrogen dioxide (NO2). With a 50% reduction in anthropogenic VOCs (AVOCs) emissions, the O3 concentration decreases across the entire geographic area, with reductions of 4%−10% in South China during winter and 8%−20% in urban areas during summer. When combining the reductions in NOx and AVOCs emissions, the ozone response in urban areas (VOC-limited) is determined by the positive effect of NOx emission reduction in winter and the negative effect of AVOCs emission reduction in summer. An exception is found in the response during summertime in South China, where the role of biogenic VOCs in ozone formation is crucial due to relatively high temperatures and the existence of vegetation surroundings.

Summertime increases in the concentration of oxidized VOCs (OVOCs), particularly aldehydes and alcohols, are attributable to the reduction in $NO_x$ emissions. This enhancement subsequently enhances the atmospheric oxidative capacity through the photolysis of OVOCs and the oxidation of alkenes by OH radicals; it favors the formation of ozone. A significant decrease in particulate nitrate and in secondary organic aerosols is derived following the reduction in $NO_x$ and AVOCs emissions, respectively. These reductions in the aerosol concentration contribute to $O_3$ formation, through enhanced levels of hydroperoxyl ($HO_2$) radicals associated with a reduced loss via aerosol uptake, and a diminished aerosol extinction. To effectively mitigate ozone pollution in urban areas, simultaneous reductions in the emission of $NO_x$ and specific VOCs species should be applied, especially regarding alkenes, aromatics, and unsaturated OVOCs, including methanol and ethanol."

**Specific Comments**

*Abstract – The authors make no mention of their conclusions related to aerosol changes.*

Author's reply: added as follows.

"A significant decrease in particulate nitrate and in secondary organic aerosols is derived following the reduction in $NO_x$ and AVOCs emissions, respectively. These reductions in the aerosol concentration contribute to $O_3$ formation, through enhanced levels of hydroperoxyl ($HO_2$) radicals associated with a reduced loss via aerosol uptake, and a diminished aerosol extinction."

*Line 258 – The authors state "A distinct increase in the surface mixing ratio of HO2 radical is derived in southern China (by up to 5 pptv or 60%; Fig. 1b). This enhancement is related to the increased mixing ratio of the OH radical found in urban areas, resulting in enhanced HO2 levels via VOCs oxidation, and a reduced HO2 loss via the aerosol uptake, as the aerosol load is reduced (see Sect. 3.2.3) (Song et al., 2021)." However, the increased HO2 appears to be present almost everywhere, while the increase in OH is highly localized to urban areas as the author's state. The authors*

*should be clearer that the broader increase in HO2 must be due to the reduced HO2 loss via aerosol uptake. If the authors tracked production from that chemical pathway, they could show a map of that rate decrease which could help. The authors could also consider a plot of the decrease in PM2.5.*

Author's reply: We agree with your suggestion for the broader impact of aerosol decrease on the enhanced levels of $HO_2$ radicals. We changed the statement with emphasis on the broader impact of aerosol uptake of $HO_2$ and shown it below (Line 337-342 in text).

"A distinct increase in the surface mixing ratio of the $HO_2$ radical is derived in South China; it reaches 5 pptv or 60% (Fig. 2e). This increase contributes to a higher ozone level through the reaction between $HO_2$ and NO. The enhancement in the urban $HO_2$ concentration results from the increased levels of the OH radical via VOCs oxidation. The reduction in the aerosol load derived in South China as a result of the reduced NOx emission is responsible for the reduced loss of $HO_2$ by aerosol uptake (see Sect. 3.2.3)."

*Figure 1&2 – I would suggest adding the 'TOTAL' case to Figure 1 and Figure 2. Figure S4b appears to be mislabeled as January instead of July.*

Author's reply: changed.

*Line 304 – What do you mean by "These changes are affected by meteorological parameters including the temperature, the water vapor abundance, and the solar radiation intensity, which affect the oxidative processes (Dai et al., 2023)." Is meteorology not the same in both simulations?*

Author's reply: The statement here is trying to compare the seasonal difference in radial's distribution. To clarify the statement, we change the description and show it below (Line 396-399 in text).

"The spatial shift in the distribution of radical changes from South China in winter to North China in summer is influenced by seasonal patterns of meteorological parameters, including temperature, water vapor abundance, and solar radiation intensity, which affect the atmospheric oxidative processes (Dai et al., 2023)."

*Line 342 – It looks like the abundance of OVOCs is reduced in "all" regions, not "most" regions.*

Author's reply: we changed the statement to all regions.

*Line 347 – Can you explain why this is? "The decrease is the most pronounced in the*

*concentration of ketones" Why does the concentration of alcohols for example not seem to change at all? Is the model budget of alcohol really dominated by BVOCs, and if so, could that really be correct?*

Author's reply: Reasons for the large changes in ketones are due to the relevant speciation for ketones are mainly from anthropogenic emissions. We added a table (Table S2) to show the speciation for specific OVOCs. For the changes in alcohols, an increase can be found in the urban areas in $NO_x$ cases in summer (Figure S9c). When AVOCs emission reduced by 50%, the alcohol concentration decreased in a large part of China (Figure S13a). Limited change in alcohol can only be found at Guangzhou sites (Figure 4g), which is due to the relatively high BVOCs concentration at the sites.

*Line 376 – Is it more effective OVOC production? Or biogenic emission of OVOCs? Maybe a budget for OVOCs (AVOC vs. BVOC vs. secondary production) would help?*

Author's reply: We added a figure of alcohol to show the potential changes in the contribution of BVOCs to OVOCs. To clarify the statement, we change the sentence to (Line 474-479 in text)

"This seasonal difference is attributable to the higher photochemical formation of OVOCs during summertime, which is favored by the higher levels of temperature, solar radiation, as well as the temperature-dependent biogenic emissions. The smaller decrease in alcohols concentration (from 1.5 ppbv in winter to 0.5 ppbv in summer; Figure S13) is also supportive to our founding, as its summertime formation is highly dependent on the photochemically reactions with BVOCs (Zhang et al., 2023)."

*Line 411 – There is still a lot of VOC-limited area. Instead of "tend to be converted", maybe say x% of VOC-limited is converted to transition or NOx-limited?*

Author's reply: we added a table (Table S3) to show the changes in percentage of grid cell of different ozone sensitivity regimes in each case, with the added statement shown in below (Line 571 in the text):

"from 68.8% in BASE case to 71.9% in NOx case".

*Line 414 – Just confirming that your HO2 uptake reaction does not produce H2O2? Does it produce H2O?*

Author's reply: we added the reaction of $HO_2$ uptake to produce $H_2O$.

*Line 420 – Against suggest pulling the 'TOTAL' case into the main text.*

Author's reply: added.

*Line 431 – Can we learn something from Guangzhou? Does Guangzhou have differences in emissions compared to the other cities that would explain why it remains VOC-limited? In the N+A and TOTAL cases, does this applies also to Shanghai? What is the difference compared to Beijing and Chengdu?*

Author's reply: The reasons for the Guangzhou sites remains in VOC-limited regimes in N+A and TOTAL conditions is due to the high biogenic emissions at the surroundings of the sites. The case for Shanghai is different from Guangzhou due to the different temperature and land cover. At the sites of Beijing and Chengdu, the changes in the ozone sensitive regimes are determined by the increased production of $H_2O_2$ due to reduced loss of $HO_2$ via aerosol uptake, as the aerosol load at these two sites is at relatively high levels. To describe the underlying reasons for the changes of ozone sensitive regimes in these four sites, we improved the explanation and shown it below. (Line 590-599 in text)

"The regimes at three urban sites, which are VOC-limited in the BASE case, are modified: the ozone sensitivity at Beijing is converted to a $NO_x$-limited case (Fig. 7i), while the sites of Shanghai (Fig. 7j) and Chengdu (Fig. 7l) are shifted towards a Transition regime. The changes in ozone sensitivity at these three city sites result from the decreased production of $HNO_3$ due to reduced $NO_2$ as well as the increased production of $H_2O_2$ due to reduced $HO_2$ loss via aerosol uptake. The Guangzhou site remains in a VOC-limited region (Fig. 7k). Reasons for this exception can be the lower aerosol load (Fig. S17) and higher temperature-dependent biogenic VOCs emissions in the location (Dai et al., 2023), as its surroundings are covered by vegetations (Zhang et al., 2023)."

*Figure 6 – Shouldn't the unit be (ppbv) not (pptv)?*

Author's reply: changed.

*Table 2 shows that in winter, NOx reduction results in ozone increase in all cities, and AVOC reduction results in ozone decrease in all cities. The N+A and TOTAL cases result in ozone increases in all cities. In summer, NOx reduction results in ozone increases in all cities while AVOC reduction results in decreases in all cities. In the N+A and TOTAL cases, ozone decreases in Beijing, Shanghai, and Chengdu, but not in Guangzhou. According to Figure 5, in July, in the N+A and TOTAL cases, Guangzhou and Shanghai remain VOC-limited while Beijing and Chengdu shift to transitional conditions. Given that you get a different picture from Table 2 vs. Figure 5 (in Table 2, Guangzhou stands out) but in Figure 5, Guangzhou and Shanghai are different from Beijing and Chengdu, it might help to start with the ozone changes in Table 2, and use your other analysis to explain those changes, rather than starting with*

*radical changes and NOx vs. VOC-limited changes.*

Author's reply: we agree with these suggestions. As we mentioned in the description of structure, we first analyzed the ozone changes at the four sites in Table 2 (shown below, Line 282-292 in text), with the supported discussion in the changes of ozone sensitivity regimes (Line 590-599 in text).

"Table 2 and Figure S5 provide quantitative information on the response of ozone to emission reduction at four urban locations (Beijing, Shanghai, Chengdu, and Guangzhou) for January and July of 2018. In winter (in January), the reduction in the emission of $NO_x$ results in ozone increases of 21.3%−33.2% in all cities, while the reduction applied to AVOCs emission results in a decrease of urban ozone levels by 2.5%−18.2%. Ozone changes in the N+A and TOTAL cases follow the ozone response found in the $NO_x$ case, with concentration increases of 7.1%−22.0% and of 10.0%−22.7%, respectively. In summer (in July), the urban ozone responses to the NOx and AVOCs cases are similar to those derived for winter conditions. The calculated ozone concentrations increase by 5.5%−17.1% in response to the reduced $NO_x$ emissions and decrease by 14.5%−22.9% in response to the reduced AVOCs emissions. In the N+A and TOTAL cases, the changes in the ozone concentration follow the response to AVOCs reductions: the ozone concentration decreases at the sites of Beijing (by 5.5% and by 7.3%), Shanghai (by 2.9% and 2.6%), and Chengdu (by 3% and 2.5%). An exception is found at the Guangzhou site, where the ozone concentration increases by 1.3% in both cases; this calls for a different role of the anthropogenic emissions regarding the ozone formation at this location."

*Figure 7 – I think it would be better if 7b was on the same scale as 7a and 7c.*

Author's reply: changed.

*Line 499 – The meaning of this is unclear "followed by effect of NO4+".*

Author's reply: we changed the statement to

"followed by the reduction in the concentration of $NO_4^+$."

*Line 515 – Cite Dai et al., 2023 here for this model bias evaluation?*

Author's reply: added.

*Line 550 – Please add some discussion of the model HO2 uptake parameterization and uncertainties in the strength of this uptake (for example, is gamma 0.2 or 0.1)? Previous studies have reduced this gamma to better fit observations (e.g., Yang et al., 2023).*

Author's reply: we added some description of the uncertainty in the parameterization of aerosol uptake of $HO_2$ and shown it below (Line 556-558 in text).

"Large uncertainties still exist in the adopted value of the uptake coefficient of $HO_2$ (considered as 0.1 in this study) (Yang et al., 2023). This affects the quantitative evaluation of the aerosol effects on the ozone levels and deserves further studies."

*Line 555 – Can you better describe the calculation of AOC? Is there an equation you can add here?*

Author's reply: we added an equation of *AOC* in the main text.

*Line 557 – There is nothing in the discussion below to support this statement: "This parameter allows us to characterize the formation process of O3 and can be used as an indicator to design mitigation policies for reducing ozone pollution." I think this comes in better in the conclusions where you describe the relative importance of different VOCs to AOC. How is this different/better than the use of OH reactivity? The conclusions mention that AOC helps you to pick out "alkenes, aromatics, and unsaturated OVOCs, especially methanol and ethanol." It would help if the identification of those VOCs were discussed in Section 3.3 and not solely placed in the conclusions.*

Author's reply: We agree with this suggestion. (1) First, we changed the sentence for the parameters after the description of the relative importance of different VOCs to AOC (Line 631-633 in main text). (2) Reasons for choosing the AOC parameters rather than OH reactivity are the AOC parameters is more relevant to our topic and the AOC changes due to emission reduction can not only represent the changes in radicals but also in the feedback of radicals' changes to ozone, as involved the OH and ozone-related reactions. (3) we added the description of changes in VOCs and shown it below (Line 465-467; Line 499-482 in Text).

"This result indicates that reducing anthropogenic emissions of aldehydes and alcohols may help offset the increase in OVOCs caused by the reduction in $NO_x$ emissions."

"Considering the increases in aldehydes and alcohols levels induced by reduced $NO_x$ emission, this result also reveals a need to reduce the primary emissions of these two OVOCs to effectively control their negative impact on ozone pollution mitigation".

*Line 640 – Better to name the specific cities and instead of 'slight' give the actual increase.*

Author's reply: we added the relevant statement and shown it below.

"i.e., by 0.5 ppbv or 1.3% at Guangzhou sites"

*Line 651 – Does ozonolysis really have a net impact of increasing ozone levels? Just need clarification here on the suggestion that the net effect is positive.*

Author's reply: The effect of ozonolysis on AOC is related to the changes in the concentration of ozone and alkenes. It is hard to answer the effect of ozonolysis on the net ozone levels based on our studies, so we change to statement to

"by an increase in the atmospheric oxidizing capacity (*AOC*) through OH-related reactions".

*Line 653 – Do you mean enhance the level of OH? Otherwise, this is a nice schematic and helpful description.*

Author's reply: We modified the $HO_x$ to OH in the main text.

*Line 704 – The author's state: "The modified code in the WRF-Chem model is available upon request to the corresponding author." The best practice now seems to be to put modified code on Zenodo.*

Author's reply: We are happy to share our code with anyone interested in our studies. Compared to simply uploading the code to Zenodo, we prefer to engage in more communication with interested individuals and share more detailed information with them.

*References*

*Yang, L. H., Jacob, D. J., Colombi, N. K., Zhai, S., Bates, K. H., Shah, V., Beaudry, E., Yantosca, R. M., Lin, H.,Brewer, J. F., Chong, H., Travis, K. R., Crawford, J. H., Lamsal, L. N., Koo, J.-H., and Kim, J.: Tropospheric NO2 vertical profiles over South Korea and their relation to oxidant chemistry: implications for geostationary satellite retrievals and the observation of NO2 diurnal variation from space, Atmos. Chem. Phys., 23, 2465–2481, https://doi.org/10.5194/acp-23-2465-2023, 2023.*

---

## Author Response (AR3)

**Response to Reviewers' Comments**

**Dear Editor,**

Thank you very much for your efforts in reviewing our submission. We have carefully revised the manuscript. Attached are the detailed point-by-point responses to the review comments. For clarity, the reviewer's comments are listed below in *black italics*, while our responses and changes in the manuscript are highlighted in blue and red, respectively.

Best regards,
Guy Brasseur and co-authors

*Line 501: NO4+ is not a compound. Please check which aerosol compound you are referring to here.*

Author's reply: Sorry for the mistake. We changed it to $NH_4^+$.

*The abstract must be 250 words or less, as per ACP author guidelines. Please reduce and make sure that it follows the formula found here: https://www.atmospheric-chemistry-and-physics.net/policies/guidelines_for_authors.html*

Author's reply: We have condensed the abstract and show it below (236 words).

Despite substantial reductions in anthropogenic emissions, ozone ($O_3$) pollution remains a severe environmental problem in urban China. These reductions affect ozone formation by altering the levels of $O_3$ precursors, intermediates and the oxidation capacity of the atmosphere. However, the underlying mechanisms driving $O_3$ changes are still not fully understood. Here, we employ a regional chemical transport model to quantify the ozone changes due to a specified emission reduction (50%) for winter and summer conditions of 2018. Our results indicate that reduction in nitrogen oxide ($NO_x$) emissions increase surface $O_3$ concentrations by 15%−33% on average across China in winter and by up to 17% in the volatile organic compounds (VOCs)-limited areas during summer. These ozone increases are associated with a reduced $NO_x$-titration effect and higher levels of OH radical. Reducing the $NO_x$ emission significantly decreases the concentration of particulate nitrate, which enhances ozone formation through increased $HO_2$ radical levels due to reduced aerosol uptake and diminished aerosol extinction. Additionally, an enhanced atmospheric oxidative capacity, driven by larger contributions from the photolysis of OVOCs and OH-related reactions, also favors urban ozone formation. With additional reductions in anthropogenic VOCs emissions, increases in summertime ozone (VOC-limited areas) can be offset by the reduced production of radicals from VOCs oxidations. To effectively mitigate ozone pollution, a simultaneous reduction in the emission of $NO_x$ and specific VOCs species should be applied, especially regarding alkenes, aromatics, and unsaturated OVOCs, including methanol and ethanol.

*Also, please review the conclusion section through the same link and provide a brief comparison and context with previous studies and caveats and limitations.*

Author's reply: We modified the final paragraph of our conclusion to meet the requirements.

Overall, in urban areas, the reduction in the surface ozone levels requires a reduction in the emissions of anthropogenic VOCs. These results are consistent with the studies of W. Wang et al (2023) and Liu et al., (2023), who stated that the priority to control ozone pollution in China should be to reduce the emissions of VOCs. Our study assumes a uniform 50% reduction in the emissions of all primary VOCs. Future work should

therefore determine which of these VOCs should be reduced as a priority to determine the most effective ozone control strategy. Our results suggest that reducing emissions of alkenes, aromatics, and unsaturated VOCs, especially methanol and ethanol, should be a priority. To develop efficient mitigation strategies that reduce AVOC emissions, more detailed investigations are needed into the reactivity of individual VOCs and their potential impact on urban ozone formation.